# Antarctic ice sheet thickness estimation using the H/V spectral ratio method with single-station seismic ambient noise

**Peng Yan**[1], **Zhiwei Li**[2], **Fei Li**[1,3*], **Yuande Yang**[1], **Weifeng Hao**[1], **Feng Bao**[2]

[1]Chinese Antarctic Center of Surveying and Mapping, Wuhan University, Wuhan 430079, China
[2]State Key Laboratory of Geodesy and Earth's Dynamics, Institute of Geodesy and Geophysics, Chinese Academy of Sciences, Wuhan 430077, China
[3]State Key Laboratory of Information Engineering in Surveying, Mapping and Remote Sensing, Wuhan University, Wuhan 430079, China

*Correspondence to*: Fei Li (fli@whu.edu.cn)

**Abstract.** We report on a successful application of the horizontal-to-vertical spectral ratio (H/V) method, generally used to investigate the subsurface velocity structures of the shallow crust, to estimate the Antarctic ice sheet thickness for the first time. Using three-component, five-day long, seismic ambient noise records gathered from more than 60 temporary seismic stations located on the Antarctic ice sheet, the ice thickness measured at each station has comparable accuracy to the Bedmap2 database. Preliminary analysis revealed that 60 out of 65 seismic stations on the ice sheet obtained clear peak frequencies (f0) related to the ice sheet thickness in the H/V spectrum. Thus, assuming that the isotropic ice layer lies atop a high velocity half-space bedrock, the ice sheet thickness can be calculated by a simple approximation formula. About half of the calculated ice sheet thickness were consistent with the Bedmap2 ice thickness values. To further improve the reliability of ice thickness measurements, two-type models were built to fit the observed H/V spectrum through non-linear inversion. The two-type models represent the isotropic structures of single- and two-layer ice sheet, and the latter depicts the non-uniform, layered characteristics of the ice sheet widely distributed in Antarctica. The inversion results suggest that the ice thicknesses derived from the two-layer ice models were in good consistence with the Bedmap2 ice thickness database, and their ice thickness differences were within 300 m at almost all stations. Our results support previous finding that the Antarctic ice sheet is stratified. Extensive data processing indicates that the time length of seismic ambient noise records can be shortened to two hours for reliable ice sheet thickness estimation using the H/V method. This study extends the application fields of the H/V method and provides an effective and independent way to measure ice sheet thickness in Antarctica.

## 1 Introduction

The Antarctic ice sheet is the largest on the Earth, covering over 98 % of Antarctic continent. As a fundamental parameter of the Antarctic ice sheet, ice sheet thickness is significant for dynamic ice sheet modeling of mass balance and sea level changes (Budd et al., 1991; Gogineni et al., 2001; Bamber et al., 2001; Hanna et al., 2013). Additionally, seismic waves become more complex when traveling through an ice sheet with thickness ranging in hundreds to thousands of meters thick. Thus, accurate ice sheet thickness is a critical metric for recognizing and denoising seismic multiples trapped inside the ice sheet when imaging crustal and mantle structures below the ice sheet (Lawrence et al., 2006; Hansen et al., 2009, 2010). Therefore, better ice sheet thickness and structures can also improve the study of the geological structure underneath the ice sheet in Antarctica.

Given the importance of Antarctic ice sheet structures, many geophysical methods, such as drilling, gravity modelling, radio echo sounding (RES), and active seismic approaches including reflection and refraction, have been used in local or regional scale ice sheet thickness investigations since the 1950s (Bentley and Ostenso, 1961; Bentley, 1964; Evans and Robin, 1966; Evans and Smith, 1969; Robin, 1972; Drewry et al., 1982; Cui et al., 2016). By studying gravitational anomalies in the ice sheet, gravimetric measurements provide an indirect way to infer the average ice thickness over a region. Active seismic and RES methods can determine the ice thickness at a much smaller area by converting the echo time of seismic and electromagnetic waves into an estimation of ice thickness. Among these methods, the active seismic and RES methods are the most widely used techniques for ice thickness measurements due to their relatively high accuracy and better spatial resolution, while gravity modelling is used as a complementary way in areas where lack direct ice thickness measurements. Using these methods (with the dominance of the RES method), abundant ice thickness data were collected over the past few decades. Compiled and gridded, these increasing data volumes were used to construct the Bedmap1 and Bedmap2 databases at a resolution of 5 km and 1km, respectively (Lythe et al., 2001; Fretwell et al., 2013). However, traditional methods for estimating ice thickness still have limitations. For example, the accuracy of the gravity method is relatively low because of its intrinsically low sensitivity of a gravimeter to the gravitational anomalies related to the ice sheet-bedrock interface. In the case of the active seismic and RES methods, they require considerable economic and logistical support to collect the data. With the rapid growth of cryo-seismology in the last one to two decades, many passive seismic methods have been applied to cryospheric research (Podolskiy and Walter, 2016; Aster and Winberry, 2017). Given that passive seismic methods can mitigate logistical problem and are relatively cost-efficient (Zhan et al., 2014; Picotti et al., 2017), it is therefore of interest to explore the feasibility of passive seismic methods to contribute additional and/or better constraints to the ice sheet structure.

Teleseismic P-wave receiver functions (PRF), as a generally used passive seismic method to determine crustal and mantle discontinuities, is also sensitive to the ice-bedrock interface and the seismic properties of ice sheets. Hansen (2010) successfully modelled ice sheet thickness beneath several stations in East Antarctica using PRF. Wittlinger and Farra (2012, 2015) investigated the anisotropy of the polar ice sheet by modelling the P-to-S

wave conversion with the negative PRF amplitude. Yan (2017) confirmed that the ice thickness results derived from PRF are consistent with the Bedmap2 ice thickness database. However, large numbers of teleseismic events are needed to perform PRF; it usually takes at least a one-year period of data collection, thus greatly limiting the application of the PRF method in harsh environments such as those found in Antarctica.

In order to improve the efficiency of ice thickness investigation, we selected the horizontal-to-vertical spectral ratio (H/V) method to determine ice thickness. As a passive and non-invasive seismic method, the H/V technique has been extensively used in seismic exploration as a tool to detect sediment thickness (Konno and Ohmachi, 1998; Ibs-von Seht and Wohlenberg, 1999; Bonnefoy-Claudet et al., 2006). Considering that the sediments and ice sheet layer are both low shear-wave velocity (Vs) layers atop the high velocity bedrock, the H/V method should be suitable for determining ice sheet thickness.

Lévêque (2010) applied the H/V method to four stations in the Dome C region of Antarctica for inferring the uppermost snow layer thickness and its corresponding ice properties to a few meters depth. Picotti (2017) recently adopted the H/V method to detect glacial ice thickness ranging from a few tens of meters to ~800 m in Italy, Switzerland, and West Antarctica. The H/V method has been validated for its reliability to measure glacial thickness comparing with the radio-echo sounding, geoelectric, and active seismic methods implemented at or near the same study sites. The great advantage of the H/V method over other approaches is that there is no need to record earthquakes or active sources, since it utilizes seismic ambient noise. Moreover, the H/V method requires only a few tens of minutes of seismic ambient noise recordings at single portable three-component seismometers. This greatly enhances efficiency and reduces cost and logistical support requirements.

Shear-wave velocity is an important parameter that controls the shear-wave impedance contrast (product of density and shear-wave velocity) at the interface between the upper and the lower layers. Since the shear-wave velocity of an ice sheet is ~1900 m s$^{-1}$, and generally much higher than a snow layer (~700 m s$^{-1}$), therefore the impedance contrast of the ice sheet-bedrock half-space is not as high as that of the snow-ice sheet layer. Moreover, the H/V spectrum may be more complicated than that of a glacier or snow layer given the complex subglacial environment since there might be subglacial lakes and sedimentary layers. In addition, the internal ice structure might affect the H/V spectrum given the variations in seismic velocities induced by changes in density, and temperature, as well as the ice crystal size and orientation of an ice sheet. Whether the H/V method can be used to estimate the ice sheet thickness or not remains an open question. Although the H/V method has been successfully applied to study snow and shallow glacial thickness (Lévêque et al., 2010; Picotti et al., 2017), to our knowledge, the H/V method has not been performed to estimate Antarctic ice sheet thickness yet. In this study, we present estimated ice thickness results from 65 stations with a typical coverage deployed on the Antarctic ice sheet to verify the feasibility of using the H/V method as an effective way to measure ice thickness.

## 2 Data and methods

### 2.1 Data

Over the past two decades, several temporary seismic arrays have been deployed in Antarctica, including the Transantarctic Mountains Seismic Experiment (TAMSEIS, 2000—2003) (Lawrence et al., 2006), the Gamburtsev Antarctic Mountains Seismic Experiment (GAMSEIS, 2007—2012) (Hansen et al., 2010), and the Polar Earth Observing Network/Antarctic Network (POLENET/ANET, 2007—2016) (Chaput et al., 2014). Despite their relatively sparse distribution compared to many dense seismic arrays on other continents, these three arrays together effectively cover East, and West Antarctica as well as the Transantarctic Mountain region (Fig. 1). In these three arrays, all stations are equipped with the Güralp CMG-3T or Nanometrics T-240 broadband sensors with a sampling rate of 25 Hz or 40 Hz. Most stations are buried 1—2 meters below the surface snow to guarantee data quality (mainly to ensure good coupling and to dampen wind noise) (Anthony et al., 2015). Equipped with solar panels and rechargeable batteries, the GAMSEIS and POLENET/ANET stations work continuously year round except the TAMSEIS, and provide abundant seismic ambient noise waveforms for the H/V processing. To investigate the effectiveness of the H/V method for ice thickness measurements and the proper time length for H/V processing, we selected seismic ambient noise records lasting about five days (an example of such raw ambient noise record is shown in supplementary Fig. S1), which is much longer than that used in usual H/V data processing (only a few minutes' records for sedimentary investigations with tens to hundreds of meters thick). In total, 65 stations deployed on the Antarctic ice sheet were used in this study.

### 2.2 Methods

The single-station H/V method, extensively used in sediment structure detection, acquires reliable sediment thickness and shear-wave velocities (Nogoshi and Igarashi, 1971; Nakamura, 1989). In this method, seismic ambient noise data are collected by a three component seismometer and the ratio between the horizontal (H) and vertical (V) Fourier spectra are calculated. The principle of the technique can be understood by assuming a low velocity sedimentary layer overlying a high velocity bedrock half-space. Due to the sharp impedance contrast at the interface between the two layers, the shear-wave energy within the sedimentary layer produces a prominent peak that can be observed in the H/V spectrum.

During the relatively long history of the H/V method, extensive field experiments and numerical simulations have been carried out to confirm the correspondence between the shear-wave resonance frequency and the H/V peak frequency. Initially Nakamura (1989) proposed that the peak frequency corresponds to the transfer function for vertically incident SH waves (a polarized shear-wave that is generated when an incident shear-wave enters in a heterogeneous medium). Using numerical simulations of ambient noise in a soil layer overlying a hard bedrock, Lachetl and Bard (1994) first showed that the peak frequency is very close to the shear-wave resonance frequency. This correspondence between the H/V peak frequency and the shear-wave resonance frequency was

later confirmed by Bard (1998), Ibs-von Seht and Wohlenberg (1999), and reasserted by Nakamura (2008). The peak in the H/V spectrum may also be followed by a trough. Konno and Ohmachi (1998) found such feature in the H/V spectrum in the case of a soft sediment layer atop a hard bedrock. As pointed out by Tuan (2011), the appearance of a trough probably suggests the overlying layer has higher Poisson's ratio (or impedance contrast) than that of the underlying layer. Despite the H/V peak frequency is commonly accepted as a proxy of the resonance frequency of a particular layer, no strong evidences support that the peak amplitude indicates the amplification factor of the site and there are some controversies about the nature of the ambient noise wavefield and its sources (Sánchez-Sesma et al., 2011). During the past few decades, two research branches were formed to interpret the ambient noise wavefield: Rayleigh wave ellipticity (Fäh et al., 2001; Wathelet et al., 2004) and the full wavefield assumptions including distributed surface sources (DSS, Lunedei and Albarello, 2009, 2010) and diffuse field assumption (DFA, Shapiro and Campillo, 2004; Sánchez-Sesma and Campillo, 2006; Sánchez-Sesma et al., 2011; García-Jerez et al., 2013, 2016).

To calculate the H/V spectrum, a specialized GEOPSY program was developed by the European SESAME team, and widely used to investigate the sediment structures (Bard and SESAME team, 2005). Then an approximation equation or H/V spectrum inversion approach can be used to derive the sedimentary layer thickness with the H/V spectrum.

Under the assumption of one-dimensional velocity subsurface conditions, in cases of homogenous and isotropic sedimentary layers over a homogenous half-space, the observed peak frequency equals the fundamental resonance frequency of the sedimentary layer. Thus, the resonance frequency of the low velocity layer is closely related to its thickness $h$ through the following relationship (Ibs-von Seht and Wohlenberg, 1999; Parolai et al., 2002; Picotti et al., 2017; Civico et al., 2017):

$$h = \frac{Vs}{4f_0} \tag{1}$$

where $Vs$ is the average shear-wave velocity of the sedimentary layer, and $f_0$ is the observed peak frequency. Provided that a correct estimate of the average shear-wave velocity of the sedimentary layer is available, its thickness can be approximately estimated.

Complicated sedimentary internal structures, including anisotropy and low velocity layers beneath stations, will affect the H/V spectrum and consequently violate the assumptions of Eq. (1). Therefore, when inferring complex subsurface structures, an inversion of the full H/V spectrum can be used to explain more accurately the observed H/V spectrum. Based on different assumptions (including Rayleigh wave ellipticity, DSS, and DFA) for the interpretation of ambient noise wavefield composition, several inversion schemes have been proposed and successfully applied to study sedimentary structures (Fäh et al., 2003; Arai and Tokimatsu, 2004; Herak, 2008; Lunedei and Albarello, 2009; Sánchez-Sesma et al., 2011). These assumptions differentiate themselves in the scheme of forward calculation of the H/V spectrum. In this study, a more recently developed H/V spectrum forward calculation and inversion method based on the DFA was employed (García-Jerez et al., 2016). The DFA

was proposed on the base of the recently stated connection between the diffuse fields and the Green's function which arises from the ambient noise interferometry theory. Under this assumption, the average spectral power ($P(\omega)$) of a diffuse field along each Cartesian axis are proportional to the imaginary part of Green's tensor components at an arbitrary point $x$ and circular frequency $\omega$ (i.e. $P_i(\omega) \propto \mathrm{Im}[G_{ii}(x;x;\omega)]$, $i = 1, 2, 3$, where 1and 2 stand for the horizontal directions and 3 denotes the vertical direction; terms with 1 and 2 in fact are equal). Thus, the H/V spectral ratio is given as:

$$HV(x;\omega) = \sqrt{\frac{P_1(x;\omega) + P_2(x;\omega)}{P_3(x;\omega)}} = \sqrt{\frac{2\,\mathrm{Im}[G_{11}(x;x;\omega)]}{\mathrm{Im}[G_{33}(x;x;\omega)]}} \tag{2}$$

In a horizontally layered structure, the contribution of both the surface wave and the body wave to the $\mathrm{Im}[G_{ii}(x;x;\omega)]$ (on the right-hand side of Eq. (2)) can be computed with provided medium properties including primary- and shear-wave velocities. The detailed formulations are not stated here as they are very complicated and on account of space limitation, but readers with interest can refer to Sánchez-Sesma (2011), García-Jerez (2016), and Lunedei and Malischewsky (2015). Thus, the Eq. (2) allows for the H/V spectrum inversion as it links the real measurements and the theoretical calculation of an H/V spectrum. In the H/V spectrum inversion procedure, model spaces are set for parameters including primary- and shear-wave velocities, mass density, and thickness of each layer. The sedimentary structures can be determined when the lowest misfit between the observed and forward calculated H/V spectrum is obtained using inversion algorithms such as Monte Carlo sampling and simulated annealing.

$$E(m) = \frac{\sum_j (HV^{obs} - HV_j^{theo}(m))^2}{\sigma^2} \tag{3}$$

Where $E(m)$ is the lowest value of the misfit in the $j$ iterations, and m represents a model that is comprised of primary- and shear-wave velocities, mass density, and thickness of each layer in each iteration. $HV^{obs}$, $HV_j^{theo}(m)$ are the observed and the $j$-th forward calculated H/V spectrum, and $\sigma$ is the standard deviation associated with the $HV^{obs}$.

The H/V method has been successfully applied in studies of sedimentary structures (Ibs-von Seht and Wohlenberg, 1999; Langston and Horton, 2014; Civico et al., 2017). However, applications in ice environments are rare. Lévêque (2010) studied the snow layer thickness and the ice properties beneath four stations in Dome C region of Antarctica using the H/V method. Picotti (2017) measured ice thickness ranging from tens of meters to 800 m of six glaciers in Italy, Switzerland and West Antarctica. However, the impedance contrast between the ice sheet layer and the overlying bedrock is not as high as that of sedimentary-bedrock and snow-ice layers. Moreover, the complex subglacial environment and internal ice structure create other technical obstacles. Thus, there have been no investigations of ice sheet thickness incorporating the H/V method for measurements or estimations.

In this study, the H/V spectra of 65 stations deployed on ice were processed by using the GEOPSY software.
Under the general assumption that the seismic properties are stable throughout the whole ice column, we
calculated the ice thickness using Eq. (1) as in most seismological applications to approximate the ice sheet as a
homogeneous layer. Meanwhile, a non-linear H/V spectrum inversion method developed by García-Jerez (2016)
was adopted to constrain the observed H/V spectrum to infer the ice structure, comprised of shear-wave velocity
and thickness.
During H/V spectrum acquisition using the GEOPSY software, we remove the transient signals (earthquakes)
from noise records with the STA/LTA technique and divide the records into 600 s length windows with an
overlap of 5 %. Time series were tapered with a 5 % cosine function, and the FFT was calculated for each
component. The spectra were smoothed with a Hanning window in a bandwidth of 0.1—2 Hz on a logarithmic
frequency scale. The spectra of the two horizontal components (NS and EW) were merged to one horizontal
component spectrum by calculating their geometric mean. The spectral ratios and corresponding standard
deviation estimates between the horizontal component and the vertical component were calculated.
Having acquired the resonance frequency of the ice sheet, we adopted Eq. (1) with a uniform average
shear-wave velocity—1900 m s$^{-1}$ of the ice layer to calculate the ice thickness. This velocity used here is
reasonable given that it is in the general range of ice Vs determined by seismic experiments (Kim et al., 2010).
Moreover, this velocity has also been widely used in previous studies (Hansen et al., 2010; Wittlinger and Farra,
2012; Ramirez et al., 2016). Keeping the velocities set, the ice thickness at each station was calculated using Eq.
214    (1).

In the H/V spectrum inversion procedures, Bedmap2 ice thicknesses were used as references to build the
initial models, as along with the related seismic elastic parameters (Fig. 2, Wittlinger and Farra, 2012; Ramirez
et al., 2016). We adopted two different models assuming the ice sheet is homogenous and inner ice stratified;
respectively, as shown in Fig. 2 to perform H/V spectrum inversion. Model A is a simple homogeneous and
isotropic ice structure with an ice layer overlying the half-space. In this model, the ice thickness varies from 0.7
to 1.3 times the Bedmap2 ice thickness for each station. Model B is constructed following Wittlinger and Farra
(2012, 2015) as a two-layer ice structure in which a low shear-wave velocity lies in the lower ice layer. In this
model, the thickness of the upper ice layer and the lower ice layer were set to occupy 60—75 and 25—40 percent
of the Bedmap2 thickness, respectively. Using the non-linear Monte Carlo method (García-Jerez et al., 2016),
we retrieved the optimum solutions for model A and B. These two solutions were best fitted to the observed H/V
spectrum.
It usually takes a few minutes to about half an hour to collect seismic ambient noise waveforms in the
investigations of sedimentary layers with thickness ranging from several tens to hundreds of meters. However,
there is no experiences for the time length of recording seismic ambient noise in the Antarctic ice sheet with
several kilometers thick. It is necessary to apply the H/V method with a much shorter recording time for seismic
ambient noise, considering the harsh environment and logistical support difficulties in Antarctica. Therefore, we
investigated the feasibility and reliability of H/V method by testing a range of noise record lengths; eight hour,
four hour, two hour, and one hour intervals were tested. The processing strategies remained the same as in H/V
spectrum acquisition except the window length was changed to 200 s when calculating the H/V spectrum using
different length noise records.

## 3   Results

In this study, the H/V spectra of 65 stations were obtained. Figure 3 displays the H/V spectra of nine stations
selected from three arrays. These examples are representative of all the results, and the remaining spectra are
presented in the supplementary Fig. S2. It is clearly shown that in almost all H/V spectra there were two or three
clear peaks in the frequency band. Generally, the largest amplitude appears at the first peak located around 0.2
Hz or below, and the second and the third peaks with lower amplitudes are located at ~0.5 and ~0.8 Hz,
respectively. Following the general interpretation principles for H/V spectra (Bard and SESAME team, 2005),
the peak frequency denoting the largest amplitude should be the resonance frequency of the ice sheet layer, while
the peaks appearing with lower amplitudes at higher frequencies may indicate the shallower impedance contrast
layers. The reasonableness of considering the first peak frequency with the largest amplitude as the resonance
frequency of the ice sheet layer was verified through approximate estimation based on Eq. (1), i.e., for station
E012, the Bedmap2 ice thickness at that location is 1050 m, so the resonance frequency according to Eq. (1)
should be 0.452 Hz (the given Vs is 1900 m s$^{-1}$), and as expected was observed (0.418±0.052 Hz) in the H/V
spectrum. However, there are exceptions such as station N108 displayed in Fig.3 whose first peak (0.177±0.014
Hz) amplitude is slightly lower than that of the following peak observed at higher frequency (1.666 Hz). At this
station however, the location of the first peak correlates with the resonance frequencies (0.194 Hz) through
approximate estimation. In addition, there are some stations that have no peak frequencies correlating with the
ice sheet thickness, despite the existence of peak frequency with strong amplitude in the frequency band. Station
ST07 seen in Fig. 3 is such a case, whose fundamental resonance frequency as calculated by Eq. (1) should be
0.191 Hz (its Bedmap2 ice thickness is 2490 m). Nevertheless, no clear peak around this expected frequency is
observed in the H/V spectrum. We therefore can group the results into three categories:
1) 42 stations with first peaks denoting the largest amplitude in the observed spectrum related to the ice sheet
resonance frequency, like the E012, E018, GM02, N148, P071, ST01, ST02 stations in Fig. 3.
2) 18 stations with first peaks with slightly lower amplitude but also related to the ice sheet resonance
frequency such as station N108.
3) Five stations without peaks correlating to the resonance frequency, such as station ST07.
Figure 4 shows the H/V spectra of stations along four profiles, together with the ice sheet and bedrock
elevation extracted from Bedmap2 database for each station. As shown in Fig. 4, although the neighboring
stations are 80 km apart for profile AA', 100 km for profile BB' and DD', and 20 km for profile CC', the shape
of the spectra are similar along each profile. Also, along each profile, the peaks associated with the ice thickness
are clear and the locations of the peaks shift towards lower or higher frequencies cohering with the variation of
the corresponding ice thickness. There are four stations (N060, ST04, ST06, ST07) along the four profiles
without peak frequencies related to their corresponding ice thicknesses. This may be caused by the bad coupling
of the seismometer with the ice surface or possibly a complicated subglacial environment, for example clear
evidence indicates the existence of sedimentary layer beneath station N060.
Having identified resonance frequencies of the ice sheet, we calculated the ice thickness using Eq. (1) with the
average shear-wave velocity—1900 m s$^{-1}$. The Equation (1) estimates together with their relative errors to the
corresponding Bedmap2 ice thickness are listed in Table 1(hereafter the ice thickness estimations derived from
the approximation Eq. (1) and H/V spectrum inversions using model A and model B are defined as Equation (1),
DFA + Model A, and DFA + Model B estimates, respectively). We projected the Equation (1) estimates and the
reference Bedmap2 ice thickness for stations along the four profiles in the upper elevation panels in Fig. 4. It is
clear that the Equation (1) estimates for some stations along the four profiles are close to the reference ice
thickness like the E012, P071, and ST01 stations, while there are large deviations at some stations such as E018,
N148, and ST02. It should be noted that the ice thickness obtained from the H/V method reflects the average ice
sheet thickness beneath each station in the scale of seismic wavelength (i.e. for a peak at frequency 0.2—1 Hz
and seismic wavelength of ~2.0 km, the spatial resolution (or footprint) is about 2—10 km).
The optimum shear-wave velocity models derived from H/V spectrum inversion are presented in Fig. 5 and
supplementary Fig. S3. The observed H/V spectrum together with the synthetic H/V spectra using the two
optimum shear-wave velocity models are plotted in Fig. 6 and shown in supplementary Fig. S4. As Fig. 6 and the
supplementary Fig. S4 show, the synthetic H/V spectra of the optimum inversion results for model A and model
B at almost all stations, both fit the observed H/V spectra in peak frequency and spectrum shape. However, the
DFA + Model A estimates deviate substantially from the Bedmap2 thickness at most stations (such as N108,
N148, GM02 and ST02 in Fig. 5), and the difference extends 1 km for some stations (Fig. 7). By contrast, the
DFA + Model B estimates are consistent with the Bedmap2 thickness as the differences between them are
mostly within 200 m. The overall DFA + Model B estimates are listed in Table 1, as well as the relative errors to
the corresponding Bedmap2 ice thickness. We also projected the DFA + Model B estimates for stations along the
four profiles in the elevation panels seen in Fig. 4. This figure depicts a good consistency between the DFA +
Model B estimates and the reference ice thickness as the ice thickness at 27 stations and 46 stations out of the 48
stations along the profiles are within 10 % and 15 % threshold of the Bedmap2 ice thickness.
The results of four different length seismic ambient noise records (1 h, 2 h, 4 h, 8 h) used to obtain H/V
spectrum are displayed in Fig. 8 (and in supplementary Fig. S5). These plots show that the shape of the spectra of
the four tested record lengths are similar to the shape determined using a record five days long. The peak
frequencies of the four different length records are all within the margin of error for the peak frequency as
determined with the record five days long. Besides, we found that the longer the ambient noise record, the more
stable the peak frequency is as there are slight shifts in the peak frequency when determined with 1 h records.
This feature is obvious for stations with thin ice (less than 2 km) such as those from stations E018 (Fig. 8), E014,
E020, E024, and E028 (shown in supplementary Fig. S5). The quality of the H/V spectrum obtained from one
hour long record for stations with thick ice (over 2 km) however, is generally consistent with that determined
with the record five days long. This consistency can also be found for all stations when the length of noise record
exceeds two hours.
**4 Discussion**
Bedmap2 ice thickness was used as reference to verify the Equation (1), DFA + Model A, and DFA + Model B
estimates since we lacked actual ice thickness as obtained from the more direct and accurate ice-core drilling,
RES and active seismic methods at or near each study site. Because of various factors contributing to the
uncertainty in the Bedmap2 database such as data coverage, basal roughness, and ice thickness measurement and
gridding error, however, the Bedmap2 ice thickness is not exactly accurate with uncertainty varying from site to
site. We obtained the uncertainty of the Bedmap2 ice thickness at each station from the grids of ice thickness
uncertainty (Fretwell et al., 2013, also, the uncertainty at our study sites can be roughly seen in supplementary
Fig. S6). A close examination of the uncertainty of the Bedmap2 ice thickness reveals that the uncertainty at 52
stations ranges from 59 m to about 200 m, and the uncertainty at 57 stations is below 300 m. As the accuracy of
the H/V method is at the same scale with the uncertainty of the Bedmap2 ice thickness at the 57 stations, the
Bedmap2 ice thicknesses are adequate to verify the results derived from the H/V method. The remaining three
stations including ST09, ST13, and ST14 are excluded for validation as the uncertainty of the reference ice
thickness at these stations reaches 1000 m.
A comparison of the DFA + Model B estimates and Bedmap2 database reveals that the differences in ice
thickness at all the 57 stations are less than 400 m; there are 34 stations whose differences are within 200 m and
48 stations within 300 m; the maximum difference was 360 m at stations GM06 and N215. The relative errors of
the DFA + Model B estimates to the corresponding Bedmap2 thickness of 23 stations, 36 stations, and 58
stations are within 5 %, 10 %, and 15 % threshold, respectively. Given that the Bedmap2 ice thickness are
associated with certain uncertainties at each station (i.e. the relative errors of the uncertainty to the Bedmap2 ice
thickness are within 10 % at 49 stations) (Fretwell et al., 2013). In this sense, we conclude that the DFA + Model
B estimates have comparable accuracy to the Bedmap2 ice thickness at the study sites.
Based on the homogenous ice sheet layer assumption, most of the Equation (1) estimates are not compatible
with Bedmap2 ice thickness (Fig. 4 and Fig. 7), as the differences at 25 stations can extend 400 m and at 10
stations are over 600 m; the maximum difference reaches 910 m at station N036. Moreover, most of the DFA +
Model A estimates based on the homogenous ice structure of model A also largely deviate from the reference
Bedmap2 thickness (Fig. 7 and supplementary Fig. S3). These large deviations cannot be attributed to the
uncertainty in the reference Bedmap2 ice thickness since they made minor contributions to the large differences.
The DFA + Model B estimates, however are in good consistence with the Bedmap2 database (Table 1). A
close examination of the DFA + Model B estimates shows that it refined the Equation (1) estimates at 47 stations
to varying degrees. As at stations E012 and N036, the Equation (1) estimates deviate from Bedmap2 at 90 m and
910 m, while the DFA + Model B estimates refine the gaps to 20 m and 320 m.
We compared our results with those found in Wittlinger and Farra (2012). Using the PRF method and a grid
search stacking technique, they found that the Antarctic ice is stratified, possibly due to the preferred orientation
of ice crystals and fine layering of soft and hard ice layers under pressure. In Fig.9, we present the ice sheet
structure for 12 stations common to both studies. It is clear that the interface separating the upper and the lower
ice sheet layers determined using the H/V method and the PRF method, is consistent for almost all stations.
The agreement of two-layer ice sheet thickness with the Bedmap2 database, and the consistency of our results
to Wittlinger and Farra (2012)`s results, as well as the large deviation of Equation (1) estimates and DFA +
Model A estimates jointly support the thesis that the two-layered ice sheet models are more reasonable than an
homogeneous ice sheet layer assumption. Moreover, the Equation (1) estimates of 28 stations were close to the
reference Bedmap2 database. This consistency, however, does not strongly support the homogenous ice sheet
layer assumption as it can be attributed to the fact that the Vs values adopted in approximate estimation was
coincidental with the average velocity of the two-layer Vs models.
The examples presented in this work clearly show that the H/V method with seismic ambient noise can be
effectively used to measure ice sheet thickness. However, there are also some limitations that may affect the
results. Shear-wave velocity (Vs), as the key parameter for H/V spectrum inversion and approximate estimation
using Eq. (1), will significantly affect the effectiveness and uncertainty of the H/V method. We can see from Fig.
6 that the synthetic H/V spectra from the optimum Vs profiles of model A and model B for the N108, GM02 and
N148 stations, match the observed H/V spectrum. The DFA + Model A estimates and the DFA + Model B
estimates at these stations however, are remarkably different as the DFA + Model B estimates are more closely
match the reference Bedmap2 ice thickness than the DFA + Model A estimates (Fig. 5). Also evident in these
results is a directly proportional relationship between ice thickness and the Vs as expected from Eq. (1) in
approximate estimation. Given a ~5 percent variation in the average shear-wave speed of the ice layer, then ice
sheet thickness estimation will result in a similar variation such as 150 m for a station with 3 km thickness.
Accurate known Vs profiles are therefore prerequisites when obtaining reliable H/V spectrum inversion results,
as well as for approximate estimations using Eq. (1).
It is evident that the longer the noise record, the more stable the observed peak frequency is as the sources of
the seismic ambient noise are more evenly distributed, spatially and temporally. This is significant for stations
with thin ice primarily due to the fact that thin ice sheet layers are excited by high-frequency waves such as
winds and other sources (Picotti et al., 2017). Thus, a longer ambient noise record can improve the stability of
the H/V spectrum. In our study, we found that the quality of the H/V spectrum is generally better for thick ice
sheet layers than for thin ice sheet such as stations E012, E018, E024, E026, and E028 with relatively smaller ice

thicknesses than other stations. The H/V spectra for these stations exhibited less stability when the lengths of noise records decreased (Fig. 8 and supplementary Fig. S5). Their peak frequencies obtained from a one hour long record slightly deviate from the peak frequency determined with a five day record. These deviations consequently could lead to uncertainties in ice thickness estimation. While for stations with thick ice, both the shape and the peak frequency determined using a one hour long record are generally consistent with those obtained from a five day long record. Given that the variation of ice thickness at the study sites (from 600 m to about 4 km), generally covers the range of the whole Antarctic ice sheet thickness, we would like to suggest a uniform record length of two hours in H/V method application in Antarctica, in terms of stability and efficiency.

## 5  Conclusions

The H/V method is proposed as a reliable, efficient method to investigate the Antarctic ice sheet thickness. The H/V method is effective for identifying the fundamental resonance frequency correlating with the ice sheet thickness. In this approach, the ambient noise recording length can be as short as 2 hours, reducing costs and increasing efficiency. Equation (1) can retrieve a fast and approximate estimation of the ice thickness but should be used with care since the shear-wave velocity varies at different sites. H/V spectrum inversion, however, unlike estimation with Eq. (1), is robust and can obtain reliable ice thickness results with given seismic properties. Moreover, the H/V spectrum inversion ice sheet thickness results are consistent with the reference Bedmap2 database. Our results also support the argument that the Antarctic ice sheet has a two-layer structure. The H/V method is an excellent approach that provides new and independent ice sheet thickness estimations. What makes this new approach most attractive are the ease and economy of seismic ambient noise waveforms collection when deploying a single seismometer for short time intervals. Finally, we hope that specific seismic experiments can obtain more accurate shear-wave velocity profiles in the ice sheet, thus making better constraints for H/V method results.

Supplementary materials include:

Figure S1, S2, S3, S4, S5, S6 in pdf format

*Competing interests.* The authors declare that they have no conflict of interest.

**Acknowledgement**

We thank the editor, Kenny Matsuoka, and two reviewers, Andreas Köhler and Adam Booth for their critical and
helpful comments and suggestions that greatly improved the manuscript. We thank Sidao Ni for helpful
discussion on the manuscript. This work was supported by the State Key Program of National Natural Science of
China under Grant 41531069, the Chinese Polar Environment Comprehensive Investigation and Assessment
Programs under Grant CHINARE2017-02-03, and the Special Funds for Basic Scientific Research of
Universities under Grant 2015644020201. Seismic data are obtained from the Incorporated Research Institutions
for Seismology (IRIS). Figures in this study were plotted using Generic Mapping Tools (GMT).

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

Table 1 Ice thickness results obtained from this study
(Error of Equation (1) estimates listed in column 4 is obtained by averaging the thickness using the resonance
frequency $f_0$ and its corresponding standard deviation $\sigma$ (i.e. $\Delta h = \left( \left( \dfrac{Vs}{f_0} - \dfrac{Vs}{f_0+\sigma} \right) + \left( \dfrac{Vs}{f_0-\sigma} - \dfrac{Vs}{f_0} \right) \right)/2$ ). The relative errors
listed in column 5 and 7 are calculated using $\dfrac{|Equation(1) - Bedmap2|}{Bedmap2}\cdot 100\%$ and $\dfrac{|DFA+ModelB - Bedmap2|}{Bedmap2}\cdot 100\%$,
respectively)

| Station | Bedmap2 (km) | Resonance freq. (Hz) | Equation (1) (km) | Relative error | DFA+Model B (km) | Relative error |
|---|---|---|---|---|---|---|
| BENN | 1.56 | 0.222±0.034 | 2.14±0.33 | 37.18% | 1.73 | 10.90% |
| BYRD | 2.16 | 0.222±0.022 | 2.14±0.21 | 0.93% | 2.33 | 7.87% |
| E012 | 1.05 | 0.418±0.052 | 1.14±0.14 | 8.57% | 1.03 | 1.90% |
| E014 | 0.66 | 0.914±0.085 | 0.52±0.05 | 21.21% | 0.60 | 9.09% |
| E018 | 1.50 | 0.222±0.028 | 2.14±0.27 | 42.67% | 1.72 | 14.67% |
| E020 | 1.75 | 0.200±0.011 | 2.38±0.13 | 36.00% | 2.01 | 14.86% |
| E024 | 1.83 | 0.200±0.019 | 2.38±0.22 | 30.05% | 2.09 | 14.21% |
| E026 | 1.40 | 0.215±0.028 | 2.2±0.29 | 57.14% | 1.61 | 15.00% |
| E028 | 1.61 | 0.188±0.032 | 2.5±0.44 | 55.28% | 1.85 | 14.91% |
| E030 | 2.02 | 0.177±0.024 | 2.68±0.37 | 32.67% | 2.32 | 14.85% |
| GM01 | 3.10 | 0.155±0.018 | 3.07±0.36 | 0.97% | 3.12 | 0.65% |
| GM02 | 2.81 | 0.159±0.014 | 2.98±0.26 | 6.05% | 2.94 | 4.63% |
| GM03 | 2.52 | 0.159±0.018 | 2.98±0.33 | 18.25% | 2.88 | 14.29% |
| GM04 | 2.80 | 0.157±0.015 | 3.02±0.29 | 7.86% | 3.08 | 10.00% |
| GM05 | 3.47 | 0.146±0.020 | 3.26±0.45 | 6.05% | 3.17 | 8.65% |
| GM06 | 3.47 | 0.150±0.015 | 3.16±0.32 | 8.93% | 3.10 | 10.66% |
| GM07 | 3.03 | 0.148±0.012 | 3.21±0.26 | 5.94% | 3.08 | 1.65% |
| JNCT | 1.19 | 0.349±0.031 | 1.36±0.12 | 14.29% | 1.26 | 5.88% |
| N020 | 1.71 | 0.222±0.021 | 2.14±0.21 | 25.15% | 1.95 | 14.04% |
| N028 | 2.06 | 0.197±0.020 | 2.41±0.25 | 16.99% | 2.24 | 8.74% |
| N036 | 2.21 | 0.152±0.020 | 3.12±0.41 | 41.18% | 2.53 | 14.48% |
| N044 | 2.21 | 0.169±0.023 | 2.81±0.39 | 27.15% | 2.51 | 13.57% |
| N052 | 2.39 | 0.152±0.022 | 3.12±0.45 | 30.54% | 2.75 | 15.06% |
| N068 | 2.87 | 0.155±0.014 | 3.07±0.28 | 6.97% | 2.98 | 3.83% |
| N076 | 2.46 | 0.172±0.014 | 2.76±0.23 | 12.20% | 2.59 | 5.28% |
| N084 | 2.47 | 0.183±0.016 | 2.60±0.23 | 5.26% | 2.59 | 4.86% |
| N092 | 2.63 | 0.175±0.016 | 2.72±0.25 | 3.42% | 2.48 | 5.70% |
| N100 | 2.68 | 0.167±0.015 | 2.85±0.26 | 6.34% | 2.68 | 0.00% |
| N108 | 2.45 | 0.177±0.014 | 2.68±0.21 | 9.39% | 2.56 | 4.49% |
| N116 | 2.50 | 0.175±0.024 | 2.72±0.39 | 8.80% | 2.46 | 1.60% |




| Station | Bedmap2 (km) | Resonance freq. (Hz) | Equation (1) (km) | Relative error | DFA+Model B (km) | Relative error |
|---|---|---|---|---|---|---|
| N124 | 2.42 | 0.185±0.019 | 2.56±0.26 | 5.79% | 2.57 | 6.20% |
| N132 | 3.24 | 0.146±0.018 | 3.26±0.40 | 0.62% | 3.07 | 5.25% |
| N140 | 2.79 | 0.162±0.022 | 2.93±0.42 | 5.02% | 2.69 | 3.58% |
| N148 | 2.9 | 0.137±0.017 | 3.46±0.44 | 19.31% | 3.20 | 10.34% |
| N156 | 2.55 | 0.194±0.016 | 2.45±0.20 | 3.92% | 2.48 | 2.75% |
| N165 | 2.81 | 0.150±0.021 | 3.16±0.44 | 12.46% | 2.95 | 4.98% |
| N173 | 2.38 | 0.185±0.017 | 2.56±0.24 | 7.56% | 2.54 | 6.72% |
| N182 | 2.42 | 0.191±0.014 | 2.49±0.19 | 2.89% | 2.54 | 4.96% |
| N190 | 3.01 | 0.144±0.017 | 3.31±0.41 | 9.97% | 3.15 | 4.65% |
| N198 | 3.32 | 0.148±0.017 | 3.21±0.38 | 3.31% | 3.30 | 0.60% |
| N206 | 2.96 | 0.159±0.022 | 2.98±0.41 | 0.68% | 2.61 | 11.82% |
| N215 | 3.48 | 0.155±0.017 | 3.07±0.33 | 11.78% | 3.12 | 10.34% |
| P061 | 3.16 | 0.135±0.018 | 3.52±0.46 | 11.39% | 3.17 | 0.63% |
| P071 | 2.3 | 0.194±0.018 | 2.45±0.23 | 6.52% | 2.18 | 5.22% |
| P080 | 2.47 | 0.188±0.018 | 2.52±0.25 | 2.02% | 2.52 | 2.02% |
| P090 | 2.34 | 0.212±0.022 | 2.24±0.23 | 4.27% | 2.09 | 10.68% |
| P116 | 2 | 0.222±0.023 | 2.14±0.22 | 7.00% | 1.93 | 3.50% |
| P124 | 1.54 | 0.314±0.033 | 1.51±0.16 | 1.95% | 1.47 | 4.55% |
| ST01 | 3.02 | 0.157±0.015 | 3.02±0.28 | 0.00% | 2.95 | 2.32% |
| ST02 | 2.12 | 0.164±0.018 | 2.89±0.32 | 36.32% | 2.43 | 14.62% |
| ST03 | 1.93 | 0.236±0.019 | 2.01±0.16 | 4.35% | 1.96 | 1.33% |
| ST08 | 2.18 | 0.152±0.016 | 3.12±0.34 | 43.12% | 2.50 | 14.68% |
| ST09 | 2.32 | 0.157±0.020 | 3.02±0.4 | 30.17% | 2.66 | 14.66% |
| ST10 | 1.23 | 0.266±0.030 | 1.79±0.21 | 45.53% | 1.51 | 22.76% |
| ST12 | 1.89 | 0.185±0.020 | 2.56±0.28 | 35.45% | 2.15 | 13.76% |
| ST13 | 1.94 | 0.167±0.018 | 2.85±0.32 | 46.91% | 2.23 | 14.95% |
| ST14 | 1.54 | 0.339±0.038 | 1.40±0.16 | 9.09% | 1.44 | 6.49% |
| SWEI | 2.84 | 0.162±0.017 | 2.93±0.31 | 3.17% | 2.93 | 3.17% |
| TIMW | 2.57 | 0.175±0.020 | 2.72±0.32 | 5.84% | 2.65 | 3.11% |
| WAIS | 3.37 | 0.127±0.015 | 3.73±0.43 | 10.68% | 3.71 | 10.09% |


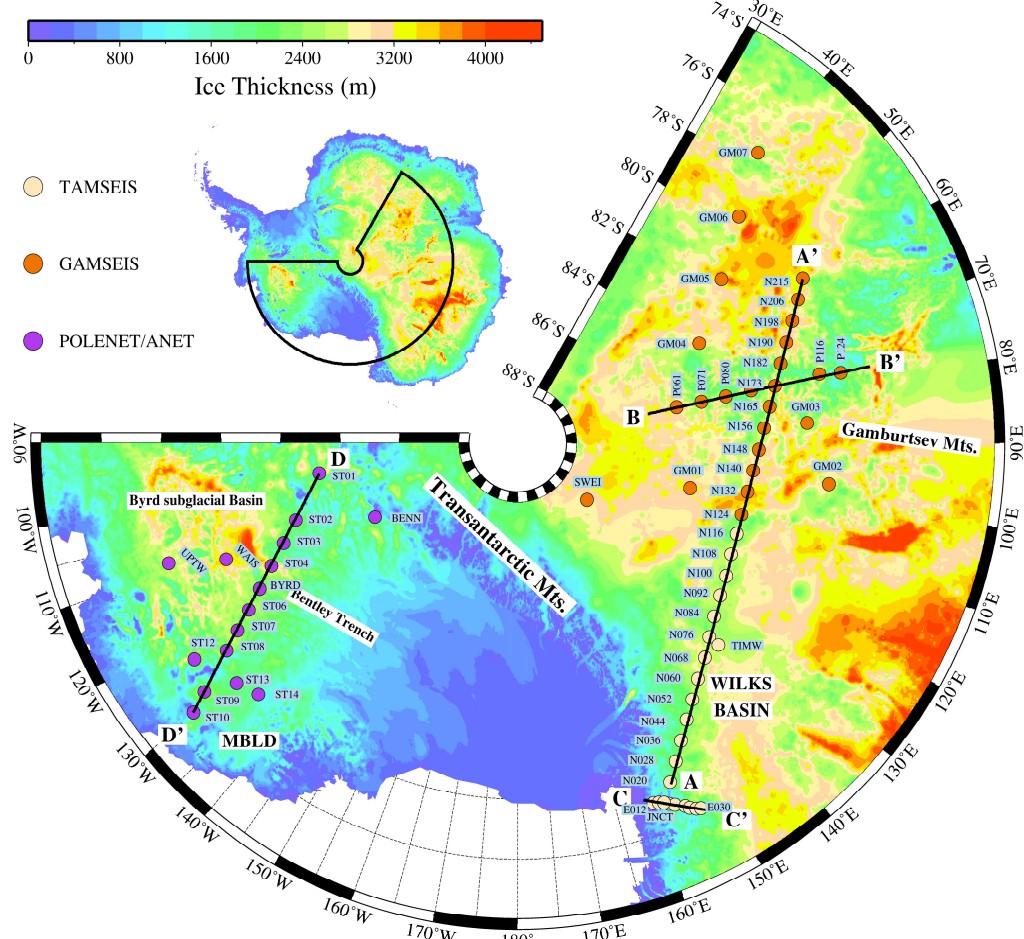


**Figure 1.** Locations of the three seismic arrays used in this study. Some stations are lined to four profiles marked with AA',

BB', CC' and DD'. TAMSEIS:TransAntarctic Mountains Seismic Experiment; GAMSEIS:Gamburtsev Antarctic

Mountains Seismic Experiment; POLENET/ANET:The Polar Earth Observing Network/Antarctic Network. Ice sheet

thickness data in this plot come from Bedmap2 database.



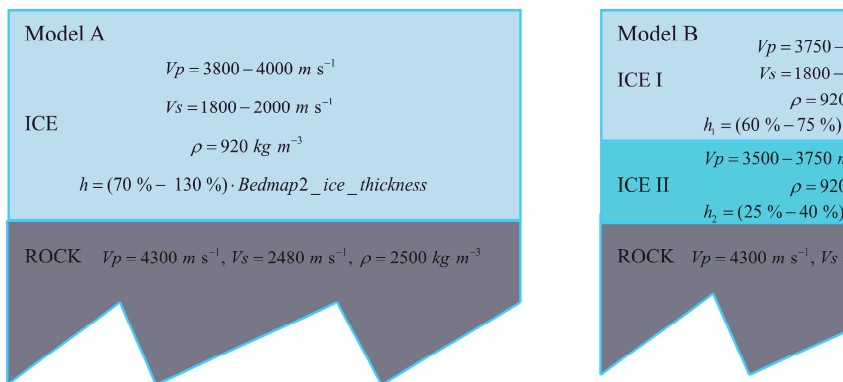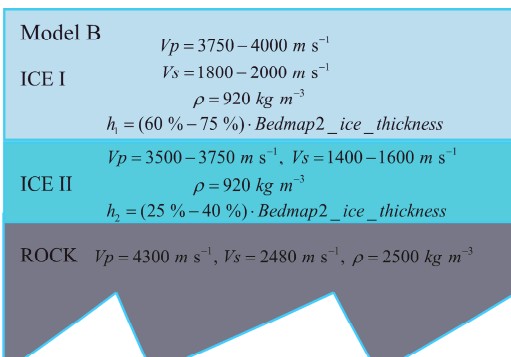


**Figure 2.** Sketches of the two ice layer models used for H/V spectrum inversion. Model A comprises a single ice layer,
while model B is a two-layer ice structure with low shear-wave velocity in the lower ice layer. The parameters used in the
two models are consistent with Wittlinger and Farra (2012).


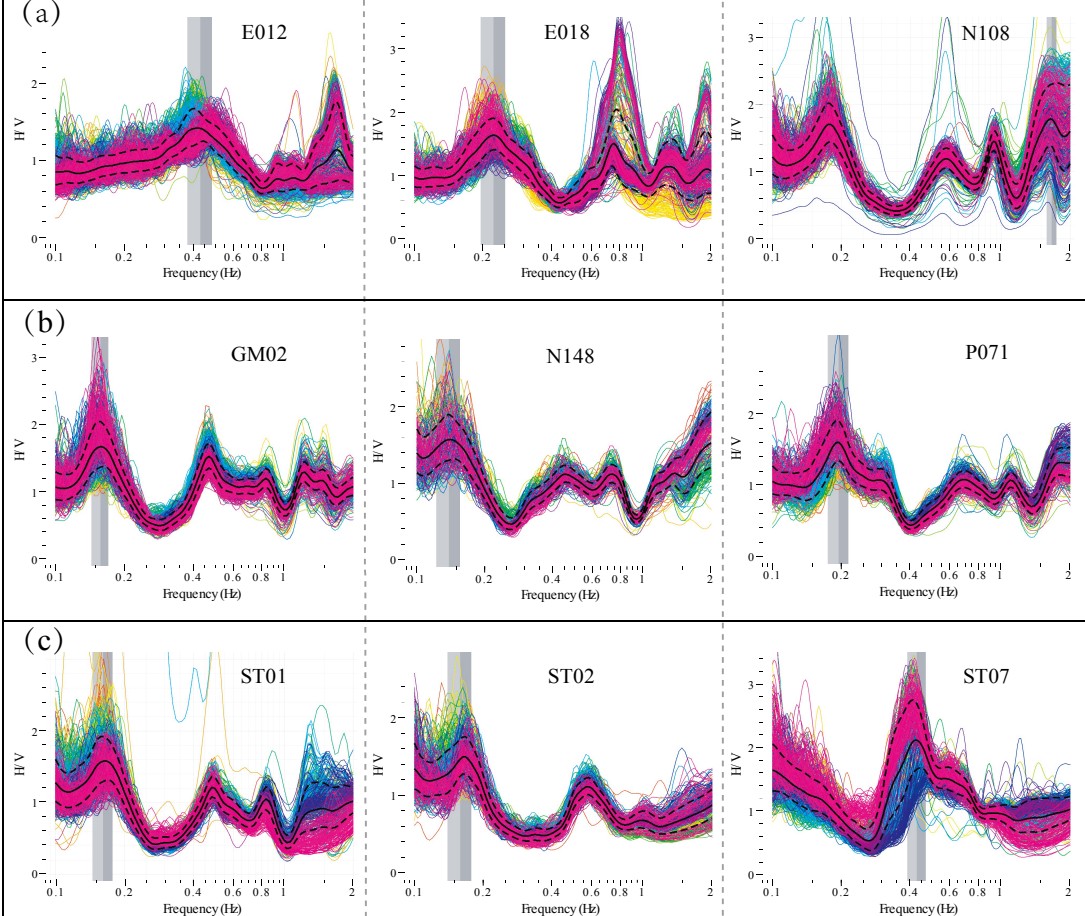


**Figure 3.** H/V spectra of nine stations shown as representative of all results in this study. Panel a, b, and c each is comprised of three stations that belong to TAMSEIS, GAMSEIS, and POLENET/ANET array, respectively (The locations can be seen in profiles displayed in Fig. 4). The H/V spectra were calculated using five-day long ambient noise record. In each spectrum, the value at the limit between the two vertical gray areas is the peak frequency, while the two gray areas denote its standard deviation. The spectra of the E012, E018, GM01, N148, P071, ST01 and ST02 stations represent 42 stations whose clear first peaks with the largest amplitudes are in agreement with the resonance frequency of the ice sheet layer. Station N108 is representative of 18 stations whose first peaks are related to the ice sheet resonance frequency but with slightly lower amplitude than peaks in higher frequencies. ST07 is the example that no peak frequency correlating to the ice thickness appears as expected in the observed H/V spectrum.

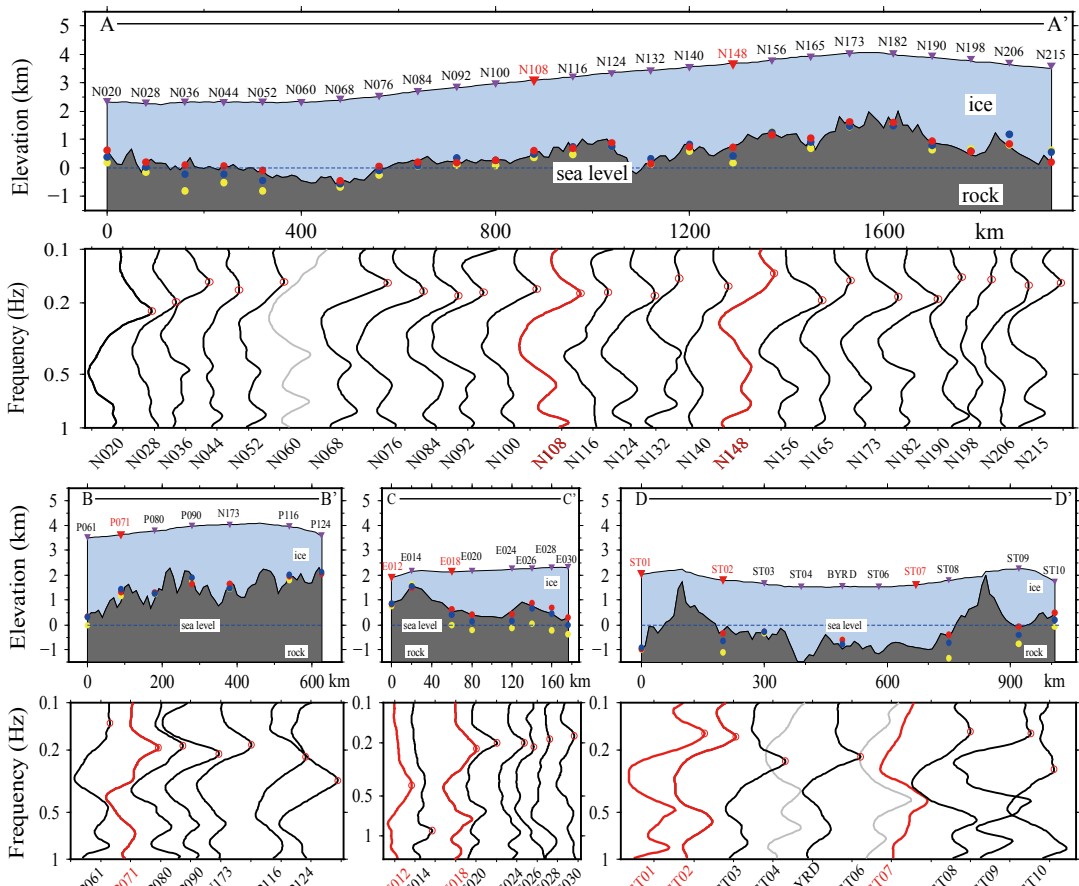


**Figure 4.** Cross section showing H/V spectra and the ice sheet thickness obtained from the H/V method at stations along
the four profiles (Fig. 1). In the below H/V spectra cross section panels, the red circles denote the resonance frequencies
correlating to the ice thickness for each station, and the spectra of the four stations without clear peaks are plotted with
gray lines. The spectra together with their station names that shown with red color, are correspondence to the stations
displayed in Fig. 3. The upper panels show the variation of the bedrock and ice surface elevation along each profile
obtained from Bedmap2 database. In these plots, the red dots indicate the reference Bedmap2 ice thickness, while the
yellow and the blue dots represent the Equation (1) estimates and the DFA + Model B estimates, respectively.


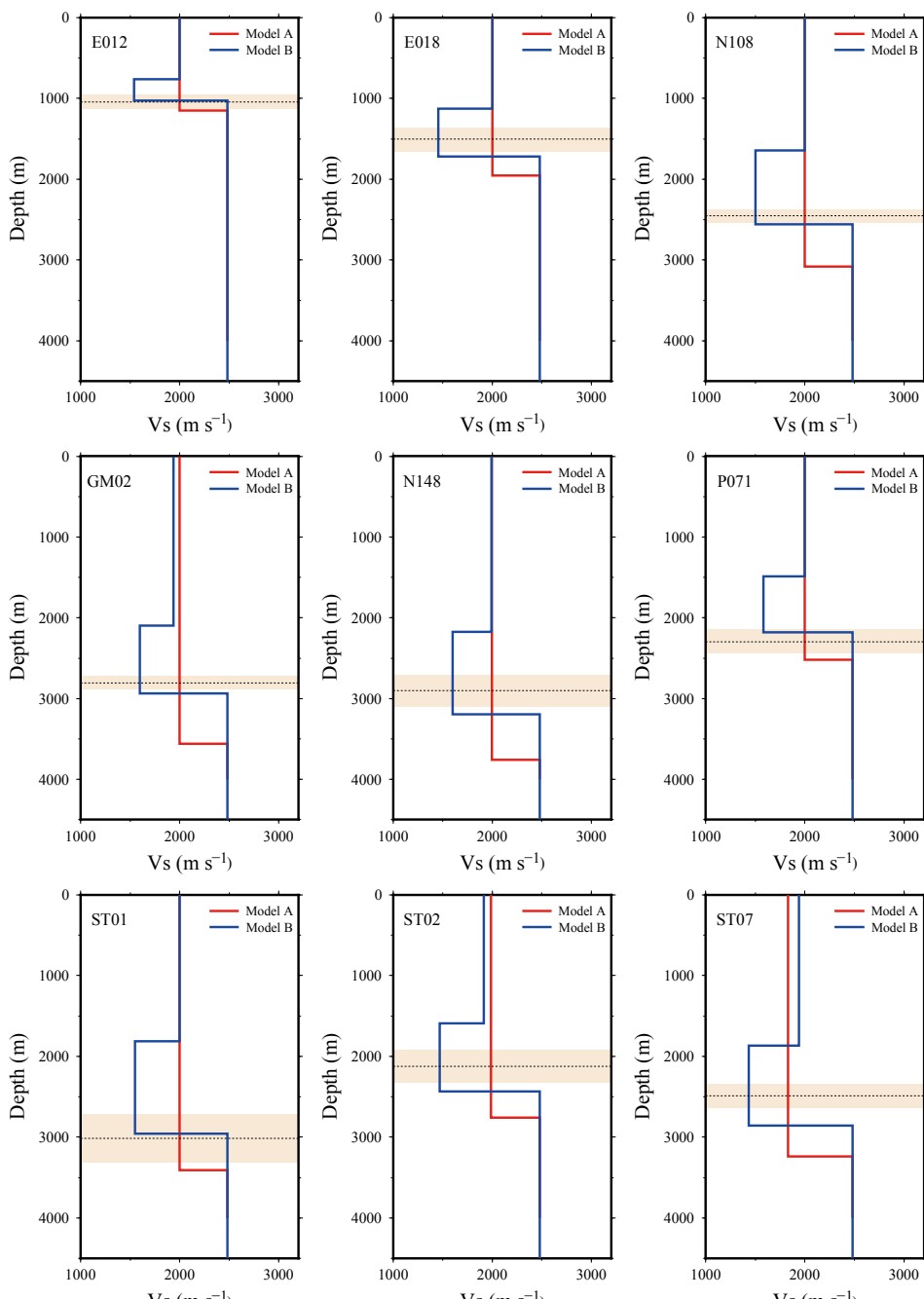


**Figure 5.** The optimum inversion shear-wave velocity models for the nine stations. The horizontal dashed line in each plot
indicates the reference Bedmap2 ice thickness, and the shaded area shows the uncertainty of the Bedmap2 ice thickness.
Apparently, the inversion ice thickness results derived from the two-layer structure (model B) are much closer to the
Bedmap2 thickness than those determined using the single ice layer (model A).



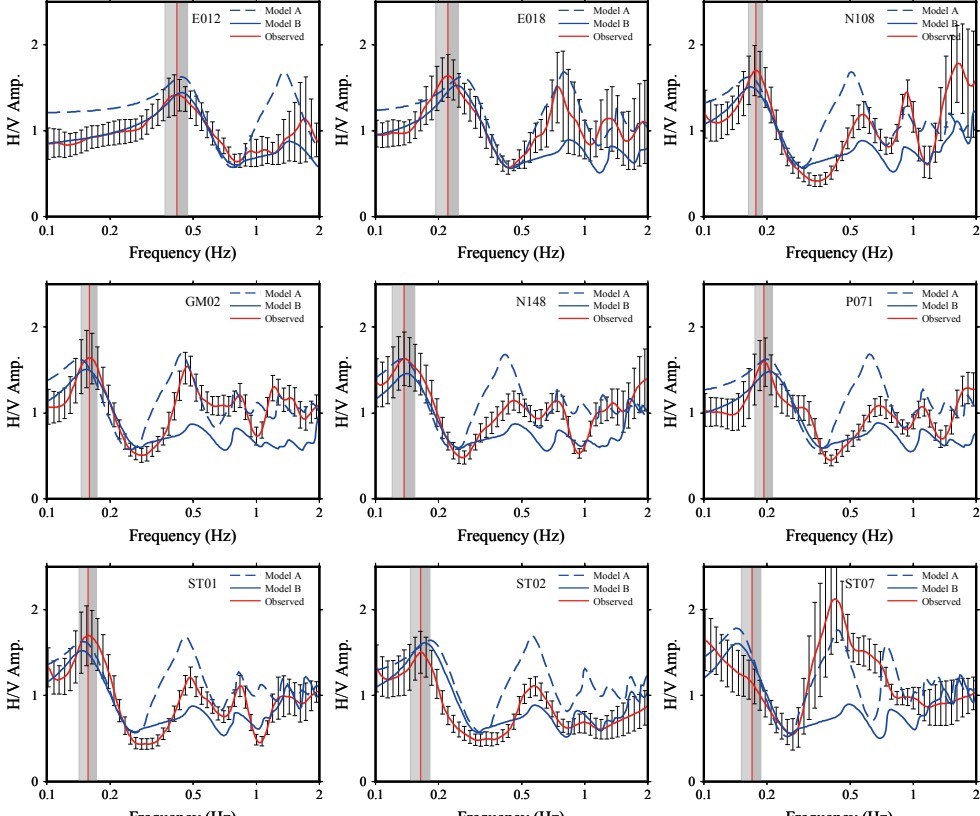


**Figure 6.** The synthetic H/V spectra and the observed H/V spectrum for the nine stations. The synthetic H/V spectra are
modelled using the optimum inversion shear-wave velocity profiles for model A and model B. In all plots except for the
last one, the vertical bars are the same as those in Fig.3 (i.e. the real peak frequency and the associated standard deviation).
As for the last one, the peak frequency is approximately calculated using Eq. (1) with its Bedmap2 ice thickness, and the
deviation is also approximated with a relative error of 10 % to its peak frequency. The two synthetic H/V spectra are both
in good agreement with the observed H/V spectrum. Note that the amplitudes of the synthetic H/V spectra are normalized
by dividing 2 in the whole frequency band.


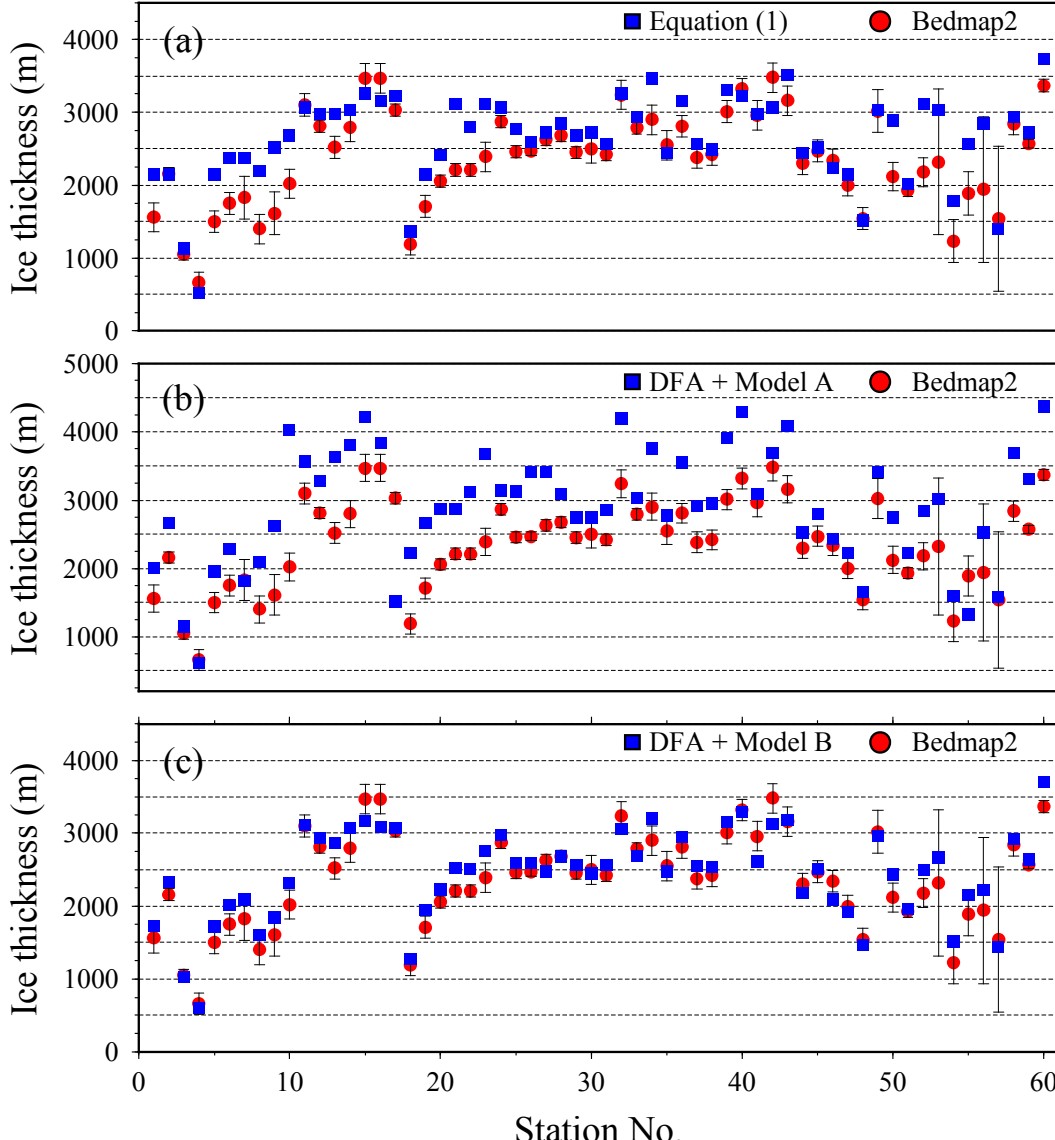

**Figure 7.** Ice thickness derived from the H/V method versus the reference Bedmap2 ice thickness. The station number of
this figure is in the same order of the stations listed in Table 1. The blue squares in panel (a), (b), and (c) represent
Equation (1), DFA + Model A, and DFA + Model B estimates, respectively. The red circles in each panel denote the
Bedmap2 ice thickness and each Bedmap2 value is marked with its corresponding error bar obtained from the uncertainty
grids (Fretwell et al., 2013).

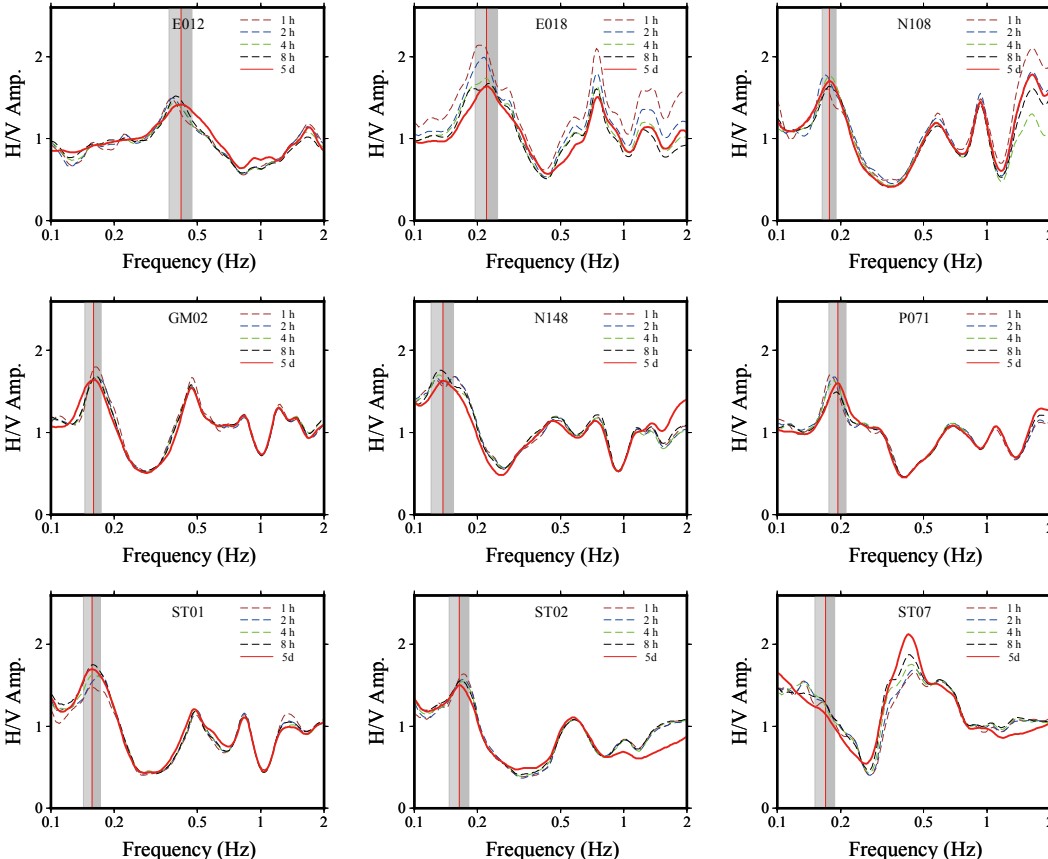

**Figure 8.** H/V spectra calculated using different lengths of ambient noise records. The vertical bars in all panels except for the last one represent the peak frequencies and the corresponding standard deviations the same as those in Fig. 3 and Fig. 6. There is a good consistence between H/V spectra determined with different tesing length of noise records (1 h, 2 h, 4 h and 8 h) and the spectrum with record five-day long, both in locations of peak frequencies and the spectra shape. However, the peak frequency obtained from 1 h record slightly deviates the peak frequency determined using 5 d record for the E012 station.

623

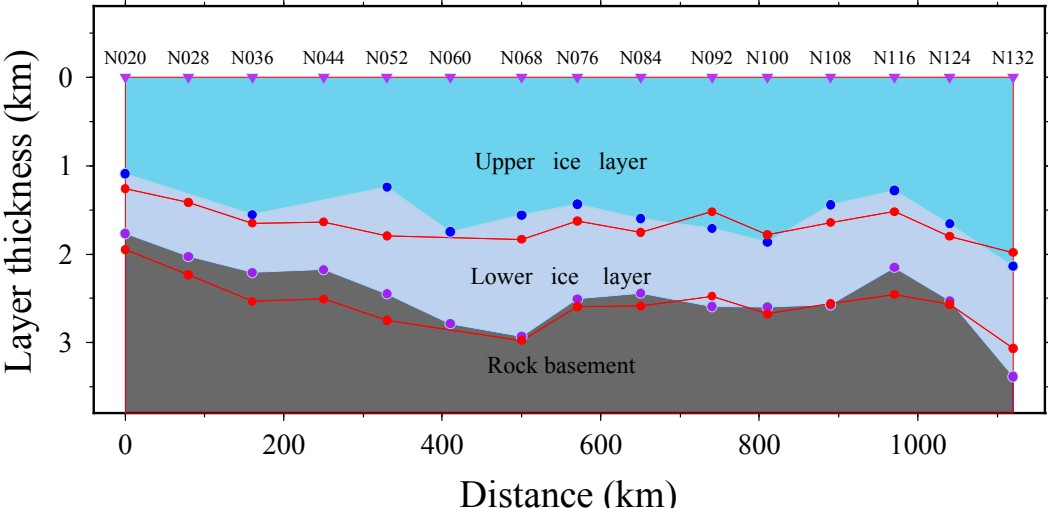

624

**Figure 9.** Comparisons of the two-layer ice sheet structure obtained from our study and Wittlinger and Farra (2012)'s. The red dots shown in shallower depth denote the interface between the upper and the lower ice sheet obtained in this study. The interface is generally consistent with that (as the blue dots shown) determined with the PRF method and a grid search stacking technique (Wittlinger and Farra, 2012, Table 1). The red dots shown in deeper depth represent the overall ice thickness derived from model B, which is also consistent with the radar ice thickness (as the purple dots shown) adopted by Wittlinger and Farra (2012).

631