# Peer review of "Antarctic ice sheet thickness estimation using the H/V"

_The Cryosphere, 2017_

## Referee Comment (RC1) · A. Köhler (Referee) · 27 Sep 2017

General comments:

The use of horizontal-to-vertical spectral ratios (H/V) is a well-established method for geophysical shallow sub-surface investigations which is mainly used within the context of seismic site-effect studies and to infer sediment depths. It has been recently applied on glaciers to infer ice thickness for the first time which showed the potential of this passive seismic method to provide complementary observations for cryospheric research. To my knowledge the H/V method has not been applied to measure ice sheets thickness before. Therefore, this study is highly appreciated. The paper is well-written

and presents conclusive and encouraging results. I have no major concerns about this manuscript, however, there are a few issues and details I would like the author to comment on and to add in the paper.

Specific comments:

(1) I suggest to briefly discuss the origin of the H/V spectra. A full discussion is beyond the scope of this study, but it would be helpful for future applications to know more about the basic assumptions and their reliability. Different contributions to the H/V amplitudes have been discussed since the emergence of this method such as SH wave resonance, Rayleigh wave ellipticity, and Love wave Airy phases. Recently, forward-modeling schemes based for example on the diffuse field theory have been proposed that take into account all seismic wave types (Jose Pina-Flores et al, 2017; Garcia-Jerez, 2016, Lunedei and Malischewsky, 2015). In the present paper this new method is used to invert the spectral ratios for the sub-surface structure. As far as I understood the code of Garcia-Jerez (2016) allows for separate computation of the contribution from different wave types. In the considered frequency band, ocean microseisms usually contribute most to the background seismic noise, so I would expect the contribution from Rayleigh wave ellipticity to the H/V spectra shape to be dominant. Is this the case here?

(2) What are the limitations of the H/V inversion method (e.g., non-uniqueness) and, most important, what are the error bars of the inverted velocity structures (please add in Figure 5)? How much is the velocity allowed to vary in the parameter space?

(3) I am also curious to what extent the other H/V peaks directly tell us something about the sub-surface structure. Can they be interpreted as multiples / overtones of the main peak, or do they correspond to other interfaces within the ice? Is there a peak or a through in the spectrum which corresponds directly to the interface within the ice that you invert for (Model B)?

In their paper, Picotti et al (2017) discuss the implication of soft-bed vs. hard-bed

sub-glacial conditions on the H/V spectra, and interpret the presence of a H/V peak or a through to be related to these conditions. Do you have any indications that the presence of sediments (soft-bed) or sub-glacial lakes lead to similar observations, i.e., a trough in the H/V spectrum that is related to the interface depth, e.g. at station N060? Is the inversion scheme you use able to take this into account? Or in other words, is the halfspace velocity allowed to become lower than the ice-sheet velocity?

(4) What is the physical model behind the two layer ice sheet model (model B)? What is the explanation for the low-velocity ice layer and are the inverted velocity values realistic? Does it make sense or have you tried to use a more complicated structure in the inversion (allow more layers and low velocity layers everywhere)? Maybe this could improve the fit even more.

(5) How is the peak frequency and it's error estimated? For example in Fig 4 the picked frequency does not seem to correspond to a maximum in the H/V spectra for stations N198 and ST07.

(6) Write some words about the spatial resolution (or footprint) of the H/V method. To what extent and where could existing ice sheet maps in Antarctica (or elsewhere) be improved using the H/V method in future seismic field experiments?

(8) Fig 6: It is unclear to me why the synthetic spectra are divided by 2. Isn't this supposed to be the best fit of the data? Then, why do the amplitudes do not match?

Technical corrections:

In references: Change "Jean-Jacues L. to "J.-J. Leveque"

---

## Referee Comment (RC2) · A. Booth (Referee) · 17 Oct 2017

**GENERAL COMMENTS**

I thought this was a good paper that applies a relatively novel method in an Antarctic environment. The paper is generally well-written, though could benefit from more quantitative discussion and consideration of its limitations. The scope of the paper matches that of The Cryosphere and, with revision, I think it will be a good addition to the literature. I make some specific comments on three main shortcomings below, then mention some smaller issues that would be required in a corrected manuscript.

[Figure]

SPECIFIC COMMENTS:

The authors show the application of the H/V seismic technique for quantifying the thickness of an Antarctic ice sheet. Two approaches are tested, based on the estimation of resonance frequencies and a more-developed inversion approach. Ice thicknesses are then compared to observed depths in Bedmap2, with the authors concluding that inversion approach is preferred but still acknowledging that some mismatch between the inversion and the Bedmap2 reference. In the paragraphs below I suggest some areas where the paper could be improved. I would emphasise that I do think the paper will make a good contribution to The Cryosphere with some attention to these issues.

1) For a paper that considers inversion and quantitative data interpretation, there's a lack of detail in the text. While I appreciate that a thorough description of the inversion approach is perhaps not required, it sits uncomfortably that there is only one simple equation in the paper – and no presentation of the raw data or the inversion approach. The authors also consider the uncertainty in Bedmap2, but give much less attention to the uncertainty in their approach (which seems counter-intuitive since I'd suggest that the uncertainty in Bedmap2 is always going to be much less than in the H/V method). Table 1 does list uncertainties in resonance frequencies, but how these are defined should be clarified. For example, peaks E012 and N148 in Figure 3 seem to be more poorly defined than others, yet their uncertainty in Table 1 seems to be consistent with the wider dataset.

The lack of uncertainty analysis sits a little uncomfortably with the frequent description of the method being "reliable" (first instance in L16) and robust. These are subjective terms that would be best qualified with numerical evidence. This is not to say that the method is unreliable, but the authors could do more to demonstrate this rather than relying on qualitative descriptions. Just present the observations and let the readership decide!

2) The authors also seem very keen to justify the need for H/V analysis, in part by

pointing out the drawbacks in other techniques (e.g., L40-96). Some of these points are valid – gravity modelling is clearly a rather low-resolution technique (although the reference to gravity data processing in L54 is very out-dated) – but I don't see that the 'economic and logistical' requirements of H/V acquisition would be significantly less than RES or seismic. The authors could lessen the criticism of these methods, and present the case for H/V analysis more simply as another interesting option for a field survey.

Additionally, the authors often point out that this is the first application of the technique on an Antarctic ice sheet: I'm also unsure that this in its own right is significant. While the logistics of an Alpine study are likely simpler than an Antarctic deployment, I would suggest that the 'seismically quiet' Antarctic – featuring simpler subglacial geometries - likely offers better-quality data than in the Alps (as mentioned in L314-5) so it should be no surprise here that promising results are obtained.

To summarise this paragraph, the justification for the authors' approach should be slightly moderated: just let the results speak in the own right, and suggest how they would complement (rather than replace) existing geophysical practice.

3) The discussion section ends with some conflicting and speculative advice for H/V compliant seismic acquisition.

In terms of the conflicting recommendation, the authors propose a desirable record length for acquiring useful H/V acquisition. In L320, the authors caution against using a record length that is only 1 hour long vs. one that is 5-days long. However, in L322-323, they suggest that a 'proper' record length of 1-2 hours would be sufficient. Firstly, the word 'proper' is misused here and it is unclear what the authors mean by this – presumably they mean "a record length suitable for reliable analysis"? But more importantly, there is an inconsistency between the recommended record lengths. I don't see how a 1-hour record would be inappropriate, but a 1-2 hour record would be fine. Additionally, in terms of the cost and logistic requirements of a deployment, if

you're going to record seismic noise for 1-2 hours, why not record for 3-4 hours?! The logistic cost is presumably the same, but you'd maybe get better data quality!

In terms of the subjectivity of this recommendation, presumably the authors have longer record lengths from their seismic stations? It should be possible to show how the estimate of ice thickness converges (?) on the Bedmap2 thickness as a function of record length, and therefore remove the subjectivity from this argument.

SMALLER CORRECTIONS:

L11: "implemented at single stations using seismic ambient noise waveforms" seems rather specific for the first line of the abstract, which is just generally about H/V methods.

L16: "reliably measured" is subjective – objectify it with some performance metrics.

L31-33: "global climate change" is misplaced here. While ice sheet thickness is important to know for sea-level rise studies, linking it here to "global climate change" is a step too far.

L34: Logical jump. The sentence starting "Moreover" likely needs a new paragraph, or a bit more development from the previous sentence.

L35: The need for accurate thickness measurements is true, but it's more likely achieved with RES than it is ever going to be with H/V analysis. Yes, there are places where RES is problematic, but the places that H/V offers better accuracy and/or precision will be few and far between. This links partly to Comment (2) that I made previously.

L41-42: What is "deep seismic sounding" as opposed to the seismic reflection and refraction methods that are already mentioned?

L45: Remove "While".

L49-51: Reference to Bedmap data seems misplaced at this point in a background

review.

L54: How big a problem would terrain corrections specifically be in Antarctica? Also, the gravity processing reference (Drewry, 1975) seems very out of date.

L59: What complement, specifically, does H/V offer to established methods?

L72: Over-selling the technique: "which suggests its powerful effectiveness ...etc". As with all techniques, there will be places where H/V is problematic.

L85: Another logical jump. Before talking specifically about the analysis parameters, you need to explain what the analysis requires.

L96: Repetition of the complementary application of H/V spectra (again without clearly explaining the complement).

L103: "relatively sparse" – spares compared to what?

L106: how does burying a station "guarantee" data quality? Presumably, you mean "to improve data signal to noise ratio"?

L124: "is not that robust" – very subjective. Defend and quantify what you mean by this. What kinds of errors result?

L157: Repetition of this point about sedimentary structure investigations.

L162: Capitalise "Geopsy" for consistency with earlier instance.

L208-209: Give the frequencies in the main text. I appreciate that they are listed in the table and in the figures, but key observations could be usefully included here.

L246: Define what you consider to be "consistent" – consistent to within what threshold?

L273-274: Again, define what you mean by "adequately constrained" – to what threshold? You could just say (e.g.) that estimates are consistent within a 5% threshold and let the readership decide if this is adequate.

L282: "inverted" rather than "inversion".

L284-287: what is it about these two stations that cause them to perform so differently?

Table 1: Could be useful to have % error, relative to the bedmap thickness?

Figure 3: Needs a colour key.

Figure 4: Plot the elevation panels at the same vertical scale. It's also a little unclear to me what the data in this figure show. If the red dots are the reference Bedmap2 thickness, how is the ice thickness defined in the panels showing the ice/rock interface? It can't be from bedmap, otherwise the red dots would coincide with this interface.

---

## Author Comment (AC1) · 16 Nov 2017

**Dear reviewer:**

We really appreciate your time and efforts put in the review of this manuscript. The constructive comments and good suggestions are really helpful to improve our manuscript greatly. Below are the comments (in black) and the corresponding responses (in blue).

**General comments:**

The use of horizontal-to-vertical spectral ratios (H/V) is a well-established method for geophysical shallow sub-surface investigations which is mainly used within the context of seismic site-effect studies and to infer sediment depths. It has been recently applied on glaciers to infer ice thickness for the first time which showed the potential of this passive seismic method to provide complementary observations for cryospheric research. To my knowledge the H/V method has not been applied to measure ice sheets thickness before. Therefore, this study is highly appreciated. The paper is well-written and presents conclusive and encouraging results. I have no major concerns about this manuscript, however, there are a few issues and details I would like the author to comment on and to add in the paper.

(1) I suggest to briefly discuss the origin of the H/V spectra. A full discussion is beyond the scope of this study, but it would be helpful for future applications to know more about the basic assumptions and their reliability. Different contributions to the H/V amplitudes have been discussed since the emergence of this method such as SH wave resonance, Rayleigh wave ellipticity, and Love wave Airy phases. Recently, forward modeling schemes based for example on the diffuse field theory have been proposed that take into account all seismic wave types (Jose Pina-Flores et al, 2017; GarciaJerez, 2016, Lunedei and Malischewsky, 2015). In the present paper this new method is used to invert the spectral ratios for the sub-surface structure. As far as I understood the code of Garcia-Jerez (2016) allows for separate computation of the contribution from different wave types. In the considered frequency band, ocean microseisms usually contribute most to the background seismic noise, so I would expect the contribution

from Rayleigh wave ellipticity to the H/V spectra shape to be dominant. Is this the case here?

Response: Thanks a lot for your helpful and constructive comments. Relative discussions were added in the revised manuscript in terms of the H/V curves interpretation and its reliability.

It is true that along the relative long history of H/V method, different seismic phases (body waves only, surface waves only, or a mix of them) were taken into account to the H/V curves interpretation and synthetic modeling. In this study, we adopted Garcia-Jerez (2016)'s method based on the DFA assumption involving both surface waves and body waves to forward calculate and invert the H/V spectral ratio. We agree that the ocean microseisms contribute most to the background seismic noise in the considered frequency band. In analysis of the contribution of different seismic waves to the H/V spectrum, it turns out that the surface wave plays a dominant role in the lower frequency part (0.1—0.3 Hz), while the body wave controls the shape of the H/V spectrum in the frequency band of 0.3—2 Hz. In particular, it seems that Love wave plays a major role around the fundamental peak frequency. However, no specific effect of the Love wave has been tested as we cannot exclude the Rayleigh wave and the body wave at the same time in the processing. Actually, despite the extensively successful applications, there are still controversies regarding the unknown ambient vibration wavefield composition and the specific contribution of a particular wave component (Langston et al, 2009; Lunedei and Malischewsky, 2015; García-Jerez et al., 2016). Specific theoretical simulation and carefully designed experiments are therefore required to decipher insightful knowledge about the debate.

Figure 1. Contribution of different seismic waves. R, L, and B represent Rayleigh wave, Love wave, and Body wave, respectively. The number 1 stands for the mode of the particular wave and 0 indicates that the particular wave was not included in the calculation, while 500 is the number of integral points of Body wave.

(2) What are the limitations of the H/V inversion method (e.g., non-uniqueness) and, most important, what are the error bars of the inverted velocity structures (please add in Figure 5)? How much is the velocity allowed to vary in the parameter space? Response: Non-uniqueness is an inherent limitation of geophysical inversion. As in the H/V spectral inversion, there is a trade-off between the shear-wave velocity (Vs) and the ice thickness, so the synthetic H/V spectrum of different Vs models (model A and model B) can both fit with the observed H/V spectrum. In this case, some other constraints such as the Bedmap2 ice thickness and reasonable Vs profiles (Wittlinger and Farra, 2012) are necessary to evaluate the inversion results.

Considering the trade-off between the Vs and the ice thickness, we cannot obtain accurate Vs and ice thickness at the same time in the H/V spectrum inversion. We therefore assumed that previous findings regarding the velocities are reasonable (the velocity structure adopted in this study is widely used in previous studies) and didn't set very large ranges for the velocities so as to provide constraint for the H/V spectrum inversion. The range of Vp is 3800—4000 m s-1 and of Vs is 1800—2000 m s-1 for model A. As for model B, the range of Vp is 3750—4000 m s-1 and 3500—3750 m s-1

in the upper and lower ice layer, and the range of Vs is  $1800-2000 \text{ m s}^{-1}$  and  $1400-1600 \text{ m s}^{-1}$  in the upper and the lower ice layer. In this sense, we therefore don't think it is necessary to add the error bars of the velocity structures in Figure 5.

(3) I am also curious to what extent the other H/V peaks directly tell us something about the sub-surface structure. Can they be interpreted as multiples / overtones of the main peak, or do they correspond to other interfaces within the ice? Is there a peak or a through in the spectrum which corresponds directly to the interface within the ice that you invert for (Model B)?

Response: According to the studies by SESAME (2004), the secondary or third peaks in high frequencies may suggest the existence of shallower impedance contrast interfaces. However, it is not easy to confirm whether it is the case or not due to the lack of information in terms of the ice sheet structure. Based on your comments and the studies by Carcione (2016), another explanation is also possible. In the case of rigid bedrock underneath the ice sheet, the following resonance frequencies has the below relationship with the fundamental resonance frequency (f0):

$$f_n = (2n+1)f_0$$
,  $n = 0, 1, 2, ..., f_0 = \frac{Vs}{4h}$

After checking the observed H/V spectra, we found that such relationship is suitable to most stations as the following secondary or third peaks are approximately three times or five times of f0.

According to your suggestion, we find that there is a trough (f1) closely followed the peak (f0) in each spectrum of model B and the ratios of f1/f0 are in the range of 1.6—2.0. However, no trough exists in each spectrum of model A. The same feature can be found in the observed H/V spectra. The trough here probably corresponds to the interface between the lower ice sheet layer and the bedrock as Tuan (2011) indicated a trough appears when the above layer has high passion ratio or the impedance contrast is high enough between the bedrock and the particular overlying layer. Thus, the existence of a tough in the observed spectrum provides additional evidence that the lower ice sheet has low Vs structure.

Figure 2. Example of a trough feature in the synthetic and observed H/V curves. A trough can be observed in the synthetic H/V curve using the optimum inversion Vs profile of model B, which is in accordance with that of the observed H/V curve. However, no trough appears in the synthetic H/V curve using the single ice layer model (model A).

In their paper, Picotti et al (2017) discuss the implication of soft-bed vs. hard-bed subglacial conditions on the H/V spectra, and interpret the presence of a H/V peak or a through to be related to these conditions. Do you have any indications that the presence of sediments (soft-bed) or sub-glacial lakes lead to similar observations, i.e., a trough in the H/V spectrum that is related to the interface depth, e.g. at station N060? Is the inversion scheme you use able to take this into account? Or in other words, is the half space velocity allowed to become lower than the ice-sheet velocity?

Response: Before conducting the H/V spectrum inversion, we modelled the synthetic H/V spectra under both assumptions of the soft over stiff medium and the stiff over soft medium as pointed out by Carcione (2016) and Picotti (2017), to fit the observed H/V spectrum. It turns out that the soft over stiff medium is more suitable to model the ice sheet-bedrock (as shown in Fig. 3). In other words, unlike the highly deformable sediments and water as found by Picotti (2017) beneath the Whillans Ice Stream, the

basal conditions beneath our study sites are probably hard bedrock. Therefore, we didn't set soft half-space in H/V spectrum inversion.

As for the station N060, we tested the influence of a 300—500 m sedimentary layer squeezed between the ice sheet layer and the hard bedrock. It turns out that the sediment slightly shifts the whole H/V spectrum to lower frequency and make the spectrum fluctuate in the frequency band of 1—2 Hz (Fig. 4). Based on your suggestion, we have further changed the half space from a hard bedrock to a soft bedrock. We find that the sediment has similar effect. However, the fundamental resonance frequency disappears and the following secondary and third peaks shift to lower frequency (Fig. 4).

Figure 3. Effect of basal conditions on the H/V spectrum. As shown in panel b, no fundamental resonance frequency correlated to the ice sheet thickness is observed in the spectrum under soft basal condition assumption (black dashed line in panel a). Under rigid basal condition assumption (blue dashed line), the fundamental frequency and the shape of the H/V spectrum are consistence with the observed H/V spectrum.

---

## Author Comment (AC2) · 16 Nov 2017

Dear reviewer:

We really appreciate your time and efforts put in the review of this manuscript. The constructive comments and good suggestions are really helpful to improve our manuscript greatly. Below are the comments (in black) and the corresponding responses (in blue).

I thought this was a good paper that applies a relatively novel method in an Antarctic environment. The paper is generally well-written, though could benefit from more quantitative discussion and consideration of its limitations. The scope of the paper matches that of The Cryosphere and, with revision, I think it will be a good addition to the literature. I make some specific comments on three main shortcomings below, then mention some smaller issues that would be required in a corrected manuscript.

SPECIFIC COMMENTS:

The authors show the application of the H/V seismic technique for quantifying the thickness of an Antarctic ice sheet. Two approaches are tested, based on the estimation of resonance frequencies and a more-developed inversion approach. Ice thicknesses are then compared to observed depths in Bedmap2, with the authors concluding that inversion approach is preferred but still acknowledging that some mismatch between the inversion and the Bedmap2 reference. In the paragraphs below I suggest some areas where the paper could be improved. I would emphasize that I do think the paper will make a good contribution to The Cryosphere with some attention to these issues.

1) For a paper that considers inversion and quantitative data interpretation, there's a lack of detail in the text. While I appreciate that a thorough description of the inversion approach is perhaps not required, it sits uncomfortably that there is only one simple equation in the paper – and no presentation of the raw data or the inversion approach.

Response:Thanks a lot for this constructive and helpful comment. We have added some texts regarding the H/V method and the inversion approach in the revised manuscript,

as well as some relevant references for providing more details for the inversion approach. Besides, an example of raw ambient noise data (a 5-day long noise record) was shown in the supplement.

The authors also consider the uncertainty in Bedmap2, but give much less attention to the uncertainty in their approach (which seems counter-intuitive since I'd suggest that the uncertainty in Bedmap2 is always going to be much less than in the H/V method). Table 1 does list uncertainties in resonance frequencies, but how these are defined should be clarified. For example, peaks E012 and N148 in Figure 3 seem to be more poorly defined than others, yet their uncertainty in Table 1 seems to be consistent with the wider dataset. The lack of uncertainty analysis sits a little uncomfortably with the frequent description of the method being "reliable" (first instance in L16) and robust. These are subjective terms that would be best qualified with numerical evidence. This is not to say that the method is unreliable, but the authors could do more to demonstrate this rather than relying on qualitative descriptions. Just present the observations and let the readership decide!

Response: We agree with your comment that quantitative discussion instead of subjective terms should be used. We first would like to state the reason why we show the uncertainty of Bedmap2 ice thickness in this study. Due to the fact that the Bedmap2 ice thickness are associated with errors that are variable, only sites with small errors (57 stations) can be used as ice thickness validations. We therefore show the uncertainty of the Bedmap2 ice thickness at each study sites. Following your very helpful suggestion, we have calculated and listed relative errors of the calculated and inversion ice thickness to the Bedmap2 ice thickness for each station in Table 1. Relevant expressions were also modified or added in the manuscript.

The GEOPSY software used in this study calculate the peak frequency and its standard deviation using all selected signal windows (i.e. in case of no windows were discarded in the noise record, a 5-day long noise record generates 720 windows with 600 s length, the GEOPSY software calculates the peak frequency for each window and then

calculate its standard deviation using all 720 windows, an example is shown in Fig. 1).
We read the peak frequency and its standard deviation from the output file. Although
the absolute uncertainty for peak E012 and N148 seems to be consistent with other
stations, the relative uncertainty to its peak frequency (E012, 12.4%; N148, 12.4%) is
larger than the other stations (GM02, 8.8%; P071, 9.3%).

[Figure]

Figure 1. The windows (each window has a length of 600 s) used for H/V processing
are colored in panel a, and each H/V curve is calculated using the corresponding
selected window (panel b). The solid black curve (in panel b) represents H/V
geometrically averaged over all used individual H/V curves.

2) The authors also seem very keen to justify the need for H/V analysis, in part by
pointing out the drawbacks in other techniques (e.g., L40-96). Some of these points are
valid – gravity modelling is clearly a rather low-resolution technique (although the
reference to gravity data processing in L54 is very out-dated) – but I don't see that the
'economic and logistical' requirements of H/V acquisition would be significantly less
than RES or seismic. The authors could lessen the criticism of these methods, and

present the case for H/V analysis more simply as another interesting option for a field survey.

Additionally, the authors often point out that this is the first application of the technique on an Antarctic ice sheet: I'm also unsure that this in its own right is significant. While the logistics of an Alpine study are likely simpler than an Antarctic deployment, I would suggest that the 'seismically quiet' Antarctic – featuring simpler subglacial geometries- likely offers better-quality data than in the Alps (as mentioned in L314-5) so it should be no surprise here that promising results are obtained.

To summarise this paragraph, the justification for the authors' approach should be slightly moderated: just let the results speak in the own right, and suggest how they would complement (rather than replace) existing geophysical practice.

Response: Thanks for this constructive comments. We have revised relevant expressions according to your suggestions. First of all, we have removed some descriptions regarding the drawbacks of other methods. Secondly, we present the H/V method as a passive seismic method that provides independent constraints to ice sheet thickness and can be used to complement existing methods in the case of the inaccessibility of the active seismic and RES methods in terms of their large logistical support requirements.

3) The discussion section ends with some conflicting and speculative advice for H/V compliant seismic acquisition. In terms of the conflicting recommendation, the authors propose a desirable record length for acquiring useful H/V acquisition. In L320, the authors caution against using a record length that is only 1 hour long vs. one that is 5-days long. However, in L322-323, they suggest that a 'proper' record length of 1-2 hours would be sufficient. Firstly, the word 'proper' is misused here and it is unclear what the authors mean by this–presumably they mean "a record length suitable for reliable analysis"? But more importantly, there is an inconsistency between the recommended record lengths. I don't see how a 1-hour record would be inappropriate, but a 1-2 hour record would be fine. Additionally, in terms of the cost and logistic requirements of a deployment, if you're going to record seismic noise for 1-2 hours,

why not record for 3-4 hours?! The logistic cost is presumably the same, but you'd maybe get better data quality! In terms of the subjectivity of this recommendation, presumably the authors have longer record lengths from their seismic stations? It should be possible to show how the estimate of ice thickness converges (?) on the Bedmap2 thickness as a function of record length, and therefore remove the subjectivity from this argument.

Response:We are sorry that we made an unclear expression here. Due to the "aseismicity" and very limited human activities in Antarctica, the quality of noise waveforms data is generally better than that found in other areas near the urban cities. We found that the shape of the spectra of the four tested record lengths (1h, 2h, 4h, 8h) are very similar to the shape determined using a record five days long. The peak frequencies of the four different length records are all within the margin of error for the peak frequency as determined with the record five days long. Thus, the ice thickness derived from Eq. (1) and H/V spectrum inversion using 1-hour long record would not result in substantial deviations from that of long records. However, we also found that the H/V spectrum exhibited less stability for thin ice sheet when the lengths of noise records decreased, which may be attributed to the interference of the high-frequency waves such as winds and other sources within short recording time intervals (Picotti et al., 2017). Such cases were found for stations BENN, E012, E018, E024, E026, and E028 (their ice thicknesses range from 500 m to 1.8 km) in this study. For these stations, two hours should be good intervals to conduct H/V processing. Therefore, we infer that two-hour long observation is better for areas with thin ice sheet (i.e. the ice thickness is less than 2 km in most places in West Antarctica). Although one-hour record can be sufficient to conduct H/V processing, we however, would like to follow your comment to advice a uniform two-hour recording interval for data acquisition in Antarctica.

SMALLER CORRECTIONS:

L11: "implemented at single stations using seismic ambient noise waveforms" seems rather specific for the first line of the abstract, which is just generally about H/V methods.

Response: We agree with your comment and have revised this sentence.

L16: "reliably measured" is subjective – objectify it with some performance metrics.

Response: Following your very helpful suggestions below, we have calculated the relative errors of the H/V results to the Bedmap2 ice thickness. It shows that the ice thickness derived from the H/V method has comparable accuracy to the Bedmap2 ice thickness. We therefore revised "reliably measured" to "has comparable accuracy to the Bedmap2 database". The detailed performance metrics were stated in the main text.

L31-33: "global climate change" is misplaced here. While ice sheet thickness is important to know for sea-level rise studies, linking it here to "global climate change" is a step too far.

Response: Thanks for this comments. We have replaced "global climate change" with "sea level change", which would be intimately connected with ice thickness.

L34: Logical jump. The sentence starting "Moreover" likely needs a new paragraph, or a bit more development from the previous sentence.

Response: "Morevoer" was modified to "Additionally".

L35: The need for accurate thickness measurements is true, but it's more likely achieved with RES than it is ever going to be with H/V analysis. Yes, there are places where RES is problematic, but the places that H/V offers better accuracy and/or precision will be few and far between. This links partly to Comment (2) that I made previously.

Response: We agree with your comments. RES method, as a very effective method for ice thickness measurements, played and will keep playing the dominant role in ice thickness investigation in Antarctica. The H/V method, as a passive seismic methods, provides independent and new constraints for ice thickness from other perspective with relatively lower cost and logistical support. Besides, we think the H/V method could be further used to infer basal properties as Picotti (2017) conducted in glacial studies.

L41-42: What is "deep seismic sounding" as opposed to the seismic reflection and refraction methods that are already mentioned?

Response: We made a mistake here and have modified "deep seismic sounding" to "drilling".

L45: Remove "While".

Response: Revised accordingly.

L49-51: Reference to Bedmap data seems misplaced at this point in a background

Response: Sorry we didn't write it clear. We refer to Bedmap data here as to state the contribution of the existing methods for obtaining abundant data.

L54: How big a problem would terrain corrections specifically be in Antarctica? Also, the gravity processing reference (Drewry, 1975) seems very out of date.

Response: We agree that in the year of 1975, the absence of high-resolution topography data may be a big problem for terrain corrections in Antarctica. We believe the recent SRTM high-resolution topography data my greatly improve the accuracy of the terrain corrections. We have deleted this expression in the manuscript.

L59: What complement, specifically, does H/V offer to established methods?

Response: The H/V method provide new constraints on ice sheet thickness with seismic ambient noise data, which we think could also provide complementary information for the strong velocity contrast at the ice-bedrock interface. We acknowledge it may confuse, so we delete this expression in the revision.

L72: Over-selling the technique: "which suggests its powerful effectiveness … etc". As with all techniques, there will be places where H/V is problematic.

Response: We agree and have removed it.

L85: Another logical jump. Before talking specifically about the analysis parameters, you need to explain what the analysis requires.

Response: Thanks for this comments. We have added some texts regarding the reason why shear-wave velocity analysis is needed in the manuscript.

L96: Repetition of the complementary application of H/V spectra (again without clearly explaining the complement).

Response: We have removed this expression in the manuscript.

L103: "relatively sparse" – spares compared to what?

Response: The distribution of the stations was relatively sparse compared to many dense arrays on the other continents where it is relatively easy to deploy seismic stations. We have added some texts in the revised manuscript to make this point clear.

L106: how does burying a station "guarantee" data quality? Presumably, you mean "to improve data signal to noise ratio"?

Response: Yes, we mean to improve the data signal to noise ratio by burying a station below surface snow since it can ensure good coupling and reduce environmental noise (such as wind). We have revised it accordingly in the manuscript.

L124: "is not that robust" – very subjective. Defend and quantify what you mean by this. What kinds of errors result?

Response: We are sorry we didn't express it clear. The peak amplitude is assumed to correspond to the site amplification factor (which of engineering interest), while no agreement has been achieved to support the statement and many studies came to conflict and even wrong results (Lunedei and Malischewsky, 2015). As we are only interested in the peak frequency in this study, we therefore don't give a detailed description about the amplitude here.

L157: Repetition of this point about sedimentary structure investigations.

Response: This sentence was removed.

L162: Capitalise "Geopsy" for consistency with earlier instance.

Response: Revised accordingly.

L208-209: Give the frequencies in the main text. I appreciate that they are listed in the table and in the figures, but key observations could be usefully included here.

Response: Revised accordingly.

L246: Define what you consider to be "consistent" – consistent to within what threshold?

Response: Thanks for this comment. Following your very helpful suggestion, we have calculated the relative error of the inversion ice thickness to the Bedmap2 ice thickness. We found that the ice thickness at 26 stations and 46 stations out of the 48 stations along the profiles are within 10 % and 15 % threshold of the Bedmap2 ice thickness. We have revised this expression accordingly.

L273-274: Again, define what you mean by "adequately constrained" – to what threshold? You could just say (e.g.) that estimates are consistent within a 5% threshold and let the readership decide if this is adequate.

Response: Thanks again for this good suggestion. We calculated the relative errors of the inversion ice thickness to the corresponding Bedmap2 thickness at each station and found that the inversion ice thickness of 22 stations, 35 stations, and 58 stations are within 5 %, 10 %, and 15 % threshold of Bedmap2 ice thickness, respectively. Considering that the Bedmap2 ice thickness is associated with certain error at each site, we then modified this "adequately constraint" expression to "comparable accuracy to the Bedmap2 ice thickness" in the manuscript.

L282: "inverted" rather than "inversion".

Response: Corrected.

L284-287: what is it about these two stations that cause them to perform so differently?

Response: Previous finding shows there are sediment with 300—500 m thick squeezed between the ice and the bedrock layers beneath station N036 (actually, there are sediment layers beneath station N020 to N060, Anandakrishnan and Winberry, 2004; Wittlinger and Farra, 2012; Frederick et al., 2016). The synthetic H/V spectrum modelling shows that the existence of sediment will shift the resonance frequency of the ice layer in the H/V spectrum, thus leading to large uncertainty of calculated ice thickness (Fig. 2).

[Figure]

Figure 2. Effect of the sediment on the location of peak frequency. The Vs profiles (panel a) show the Vs structures with and without a 300 m thick sedimentary layer squeezed between the ice sheet and the bedrock layer. The corresponding H/V curve calculated using each Vs model is shown in panel b.

Table 1: Could be useful to have % error, relative to the bedmap thickness?

Response: Thanks for this helpful suggestion, we have revised it accordingly.

Figure 3: Needs a colour key.

Response: The GEOPSY software provides no colour key in the H/V spectrum calculation procedures. In fact, each colour corresponds to a signal windows used for computing the H/V spectrum (i.e. as a 5-day long noise record is divided into windows of 600 s length, the number of windows is 720 and there are 720 colours matching with the 720 H/V spectra, an example is shown in Fig. 1). As some windows were discarded due to transient signals (earthquakes) and some other high frequency signals, the number of windows (colours) used to compute (represent) the H/V spectrum varies for each station.

Figure 4: Plot the elevation panels at the same vertical scale. It's also a little unclear to me what the data in this figure show. If the red dots are the reference Bedmap2 thickness, how is the ice thickness defined in the panels showing the ice/rock interface? It can't be from bedmap, otherwise the red dots would coincide with this interface.

Response: We have tried to plot the elevation panels at the same vertical scale. The figure, however was not as satisfactory as it currently shows since the range of elevations largely varies in different profiles (i.e. the uniform elevation scale to plot the four panels should be 8 km, while the scale for CC' profile is only 4 km).

The elevation data along each profile were extracted using the geographical coordinates of the start and the end stations. We apologize that we made a mistake when extracting the AA profile elevation data by using a wrong longitude value of station N215 and have confused the colors marking the inversion thickness and the Bedmap2 thickness in profile AA and DD panels. This figure was corrected accordingly in the manuscript. Besides, due to the fact that some station sites are not exactly in the straight line defined using the geographical coordinates of the start and the end stations, some red dots still don't exactly coincide with the interface.

References:

Anandakrishnan, S., and Winberry, J. P.: Antarctic subglacial sedimentary layer thickness from receiver function analysis. Global and Planetary Change, 42(1), 167–176, 2004.

Frederick, B. C., Young, D. A., Blankenship, D. D., Richter, T. G., Kempf, S. D., Ferraccioli, F., and Siegert, M. J.: Distribution of subglacial sediments across the Wilkes Subglacial Basin, East Antarctica. Journal of Geophysical Research: Earth Surface, 121(4), 790–813, 2016.

Picotti, S., Francese, R., Giorgi, M., Pettenati, F., and Carcione, J. M.: Estimation of glacier thicknesses and basal properties using the horizontal-to-vertical component spectral ratio (HVSR) technique from passive seismic data. Journal of Glaciology, 63(238), 229–248, 2017.

---

## Author Response (AR2)

Dear editor and reviewer,

We appreciate your helpful and constructive comments that improved our manuscript. We have made a point-by-point reply to the comments and revised all editorial corrections suggested from you and Dr. Booth. The original comments are shown below in black, while the responses to the comments follow in blue. A marked-up manuscript is attached at the end of this reply.

**Reviews from Adam Booth**

Review of revised manuscript, "Antarctic ice sheet thickness estimation using the H/V spectral ratio method with single-station seismic ambient noise" (Yan et al.)

I thank the authors for their careful response to my first review. I'm glad to see that my comments were considered and pleased that the authors found them useful.

There is one point from my previous comment that I think the authors might have misunderstood, partly due to a poor choice of words on my part. When I commented "How big a problem would terrain corrections specifically be in Antarctica?", referring to a L54 of the paper, I was suggesting that they would unlikely be a problem over ice sheets since they are rather flat. Therefore, even if high-resolution topography data were unavailable, the terrain correction would not be significantly impeded. However, I do agree with the authors correction anyway, that high-resolution data will benefit the terrain correction, so I'm not sure any amendment is actually needed.

All of my remaining comments are straightforward – I don't anticipate that any of them would represent a particularly significant stumbling block.

1. Abstract: "In this study" is not required. *(Line 13)*
2. L58 "…and are relatively cost-efficient" *(Line 59)*
3. L76 "properties to a few metres depth" *(Line 77)*
4. L129-130. Change the sentence to "The peak in the H/V spectrum may also be followed by a trough." *(Line 132)*
5. L133-136. Unclear what is meant in this sentence; check the grammar. *(Line 135-137)*
6. L152 – lower case 'w' in 'where'. *(Line 156)*
7. L155. Is anisotropy really meant to be a case of a complicated "sedimentary structure"? Do you just mean the anisotropy of the ice?
8. L181-2. "such as studies of thickness and shear-wave velocities." This can be removed; you have already established that these are the key inferences from the H/V method. *(Line 195-196)*
9. L264 – "identified the resonance frequency", or (probably better) "identified resonance frequencies." *(Line 279)*
10. L295 – "in consistence" should be "consistent". *(Line 314)*
11. L299 – "thickness was used". (Line 318)
12. L327 – should be "inverted" rather than "inversion"? *(Line 347)*
13. 344-5 – "can be effectively used"? *(Line 368)*
14. 353 – missing character; should it be "~" *(Lin3 377)*
15. L374 – is "obtaining abundant results" necessary to include? Just say that there have been many methods applied. *(Line 397-400)*

16. Figure 2 caption. "…are referred to Wittlinger (2012)." should be "…are consistent with Wittlinger (2012)." *(Line 593)*

Response: As all Dr. Booth's comments except for the 7th one are straightforward and very reasonable, we therefore just revised each of them accordingly. These revisions are indicated by the line numbers *(the number refers to this marked-up manuscript and it is the same hereafter)*. As for the 7th comment, sedimentary layers are generally assumed as vertical transverse isotropy (VTI) medium in depth imaging study, which will also affect the shear-wave velocity.

**Reviews from editor:**

Dear authors,

I received a second review from Dr. Booth. He is overall positive and made some editorial suggestions. I assessed the revised manuscript by myself as well. Revision is significant and this manuscript can be acceptable with additional revisions suggested below. Please provide a mark-up manuscript and point-to-point responses in which corresponding revisions are clearly indicated by line numbers or by pasting the revised text.

1. Expansion of the methods section 2.2 improved the manuscript significantly. The general description at lines 138-160 (in the marked up manuscript) is satisfactory. However, the revision at lines 177-200 should be improved, though I don't expect the full description of the method here. It is helpful if you mention how HV in Equation (2) and HV^(theo) in Equation (3) are connected in a clearer way, first of all. Sigma and m are not defined in the text. Are P1, P2, and P3 the average energy densities of a diffuse field? If so, define P more explicitly. Do you need to connect P and Green's tensor components using this equation? If so, explanation on its physical meaning is helpful.

Response: As extensive studies indicated, the Green's function links the H/V spectrum and the medium properties as the imaginary part of the Green's function is proportional to the average spectral power (or average energy density) of the ambient-vibration ground-motion. In a horizontally layered structure, the contribution of both the surface wave and the body wave to the Green's function can be computed with provided medium properties including primary- and shear-wave velocities. Thus, the theoretical H/V spectrum can be calculated with known medium properties. We have revised this part to make it clear. Specific revisions can be seen in *Line 171-185 and 191-194*.

2. Define three different H/V estimates of ice thickness more clearly, namely Equation (1), DFA + Model A, and DFA + Model B (or even define symbols). To a certain point in this manuscript, I thought that Model A is used with Equation (1), not with DFA. These estimates are stated many times but in slightly different ways and thus confusing. It's better to clearly define them once and stick with these terms.

Response: Thanks for your suggestion. The ice thickness estimations obtained from the approximation Eq. (1) and H/V spectrum inversions using model A and model B are defined as Equation (1), DFA + Model A, and DFA + Model B estimates, respectively. The citations of them have been revised thoroughly in the main texts, as well as in Table 1 and Fig. 7.

3. I am not convinced yet how independent ice-thickness estimates can be retrieved using the H/V method, because BEDMAP2 ice thicknesses are used to constrain the method (lines 229-230). What happens, if there is not such constraint? To rigorously argue this independence, it is necessary to make the H/V estimate without BEDMAP2's guidance and compare these results with BEDMAP2's ice thickness.

Response: Thanks for your constructive comment. We would like to state that the layer thickness and the shear-wave are not decoupled for the H/V method as they jointly influence the H/V spectrum. Given that the shear-wave structure of the ice sheet is clear (the average Vs is ~ 2.0 km s-1), the H/V method can be used to investigate ice thickness.

Similar with the RES method that exploits two-way echo time and the electromagnetic velocity in ice, the H/V estimation using the approximation Eq. (1) is also independent. If a well-established relationship between ice sheet thickness and the shear-wave velocity has been built like that in sedimentary layer depth imaging studies, the equation alone can be used to estimate ice thickness.

As for the H/V spectrum inversion, the involvement of BEDMAP2 ice thickness can reduce the independence of its estimations. Due to the non-uniqueness of geophysical inversion, it is very helpful and sometimes necessary in some cases to exploit prior constraints to reduce the non-uniqueness. We therefore used BEDMAP2 ice thickness as references to build model spaces to conduct the H/V spectrum inversion. To respond to your comment, we think the approximate estimates obtained from Eq. (1) (that are independent from BEDMAP2 ice thickness) can also be used to serve as reference thicknesses to constrain model spaces and obtain similar results.

Rough estimation: I am not a native English speaker but I think that "approximate estimate" or "first-order estimate" can show the nature of this estimate much clearer than "rough estimate". Maybe this is a personal taste so I leave it with the authors.

Response: Yes, we agree with your comment that "approximate estimate" is much proper. We have revised it in the texts thoroughly.

Wittlinger (2012) and Wittlinger (2015): are they Wittlinger and Farra?

Response: Yes, both Wittlinger and Farra are authors of the two literatures. We have corrected the citations in the main texts.

L141: Define SH.

Response: SH, together with SV wave, are two polarizations of shear-wave (S wave) when an incident S wave enters in a anisotropy medium. We have added some texts in the manuscript to make it clear *(Line 127)*

Lines 143-144: Revise the sentence.

Response: We have revised it to make it understandable. *(Line 130)*

Lines 188-189: check sets of parentheses and brackets to show the imaginary components.

Response: We have corrected sets of parentheses and brackets. *(Line 171,174)*

L261ff: what does this error stand for? It is said at L285-286 "Error of Thickness I is associated with BEDMAP2's uncertainty". Is the error of the resonance frequency here also related to BEDMAP2's error or relevant to the quality of the resonance peak?

Response: The error stated here at L261 refers to the peak frequency standard deviation that calculated using the GEOPSY software. The mentioned "relative error of Thickness I to BEDMAP2 ice thickness" at L285-286 (listed in column 5 in Table 1) were calculated with the equation: $\frac{|ThicknessI - Bed2|}{Bed2} \cdot 100\%$ .

Error of Thickness I ( $\Delta h$ , listed in column 4 in Table 1) are obtained using Eq. (1) by averaging the calculated ice thickness using the peak frequency $f_0$ and its corresponding standard deviation $\sigma$ (i.e.

$$\Delta h = \left( \left( \frac{Vs}{f_0} - \frac{Vs}{f_0 + \sigma} \right) + \left( \frac{Vs}{f_0 - \sigma} - \frac{Vs}{f_0} \right) \right) / 2$$ ). We have clarified them in the Table 1 caption. *(Line 574-578)*

Line 262: Fig. 3, not Fig. 2.

Response: Revised accordingly **(Line 256)**

Line 351: cite Table 1.

Response: Table 1 has been cited her**e. *(Line 348)***

Line 373: remove "(Fig. 5)"?

Response: Removed accordingly. *(Line 372)*

Lines 400-404: Do you really need this paragraph? I feel that this conclusion section can be stronger if you start with the second paragraph.

Response: We agree with your comment and have removed this paragraph as this paragraph is somewhat repetitive to the introduction part. *(Line 394-400)*

Table 1 caption: Please clarify thickness I and II in a better way. See the main point #2 above. Also, explain the error shown for Thickness I in the caption briefly.

Response: We have change thickness I and II to Equation (1) and DFA + Model B according to your good suggestion. The error shown in column 4 and the relative errors listed in column 5 and 7 are clarified in the caption. *(Line 575-578)*

Figure 3: change the order of individual panels or add lines between panels to group these 9 panels into 3 sub-groups. Explain what the vertical bars mean (thick gray and thin red).

Response: We have grouped the nine stations into three sub-groups that each of them belongs to one of the three seismic arrays. The vertical bar (i.e. the two vertical gray areas) stands for the peak frequency deviation domains that calculated using all the spectra with the GEOPSY software. The left gray area seems larger than the right one as they are shown on a logarithmic frequency scale. Their values are exactly the same. We have explained it in the caption. *(Line 598-602)*

Figure 4: scale all four profiles. Now the profile A-A' takes the full page width (2 columns) for 2000 km, whereas the profile D-D' of 1000-km long takes only about one third of the page width. The use of the same scale is important because the H/V method can be more useful to estimate ice thickness when the topography is less variable. Please make sure that all fonts are larger than 8 pt (feel free to use full page width and height). Separately I think that it is more appropriate to show the four stations without corresponding peaks in gray, not in red, and highlight the nine representative stations presented in Fig. 3, 5, 6 (e.g. different color or thicker curve).

Response: We have modified this figure according to your suggestion. The profiles are plotted in the horizontal scale that is proportional to the actual length of the profiles, and in the vertical that is in correspondence with their elevations. However, due to the fact that the profile CC' is only 180 km long, no details can be seen when plot using the same horizontal scale. We therefore only shorten this profile to a certain extent. The other suggestions have also been revised accordingly. *(Line 612-616)*

Figures 6 and 8: Explain red and thin dotted vertical lines. If it shows what the thick brown bars in Fig. 3 show, consider using the same legends all in Figs. 3, 6, and 8.

Response: Yes, the red and thin doted vertical lines stand for the observed peak frequency and its corresponding standard errors calculated using the GEOPSY software, respectively. However, there are five stations (ST07 here and ST04, ST06, N060, and UPTW in supplementary Fig. S4 and S5) that are in absence of peak frequencies in the observed H/V spectrum, so we calculated the expected peak frequency using its BEDMAP2 ice thickness and Eq. (1). We also assumed a 10% standard deviation for their peak frequencies. Following your helpful suggestion, we have replaced Fig. 6 and 8 (and supplementary Fig. S4 and S5) with the revised ones using similar legends with that of Fig. 3.

Figure 7: Do you really need this figure, or can you show the same information by adding the Model A estimates to Figure 4? Figure 4 already show BEDMAP2, Equation (1), and Model B estimates. If you judge that this figure is necessary, please provide justification (would the figure be too busy to read details if all info is shown in Fig. 4?). Do you really need three panels for Fig. 7? You can show BEDMAP2 using mask, and three different estimates using markers (but then such figure is very similar to the current Fig. 4). In any case, clarify how the station number is given (is it in the same order listed in Table 1?).

Response: We have tried to show all BEDMAP2, Equation (1), Model A, and Model B ice thickness estimates in Figure 4. The figure however, as you predicted, was too busy to read details since some dots that denoting particular estimates would be masked by the other dots. Additionally, only 46 stations that obtained ice thickness estimates are showing along these four profiles. We therefore think Figure 7 is necessary to show all 60 stations on one hand and in a clearer way on the other hand. We have adjusted the order to show the three panels (now in the order of Equation (1), DFA + Model A, and DFA + Model B). Besides, the station number of this figure is in the same order of the stations listed in Table 1. We have clarified it in the caption. *(Line 640)*

Figure 9: I think that this figure can easily cause misleading, because blue dots perfectly match with the interfaces and thus Wittlinger and Farra make better estimates than this study. I recommend using BEDMAP2 thickness to mask the bed and ice.

Response: The blue dots (now purple dots) separating the interface between the ice and the bed in fact is radar ice thicknesses. Wittlinger and Farra (2012, 2015) used radar ice thickness to investigate the shear-wave velocity structure in an ice sheet. Analyzing the P-wave receiver function waveforms, they found negative amplitude in the waveforms and attributed this negative amplitude to the existence of a low shear-wave velocity layer in an ice sheet. They then investigate the shear- wave velocity structure and determined the upper and the lower ice sheet thickness using a grid search technique. This figure is plotted to show the consistence of the interface separating the upper and the lower ice layer between our results and that of Wittlinger and Farra (2012). Besides, the overall ice thickness obtained from the H/V method is generally consistent with the radar ice thickness they adopted. Following your helpful suggestion, we have changed the color of the interface to avoid misleading and clarified it in the caption. *(Line 659-664)*

Clarification for a mistake in Table 1

After checking this manuscript, we apologize that we paste the wrong values of Bedmap2 ice thickness and the DFA + Model B estimate of station ST03 in Table 1. The current listed values belong to station ST07 that was not listed here. We have corrected it in Table 1 and updated all corresponding texts and figures.

[revised manuscript text omitted]